# Beneath the Surface: Exposing and Mitigating Surface Learning in Large Language Models

## Abstract

As Large Language Models (LLMs) continue to evolve, assessing their genuine comprehension of underlying knowledge is crucial to ensure the reliability in real-world applications. To evaluate what LLMs learn, we first introduce **ME**-Test suite, including **M**athematical and **E**nglish grammar examinations, where each question is equipped with relevant knowledge to guide the model. Building upon this, we construct a sequence of questions with increasing difficulty based on Cognitive Load theory, enabling the model to perform continuous problem-solving using the dialogue history. Through a comprehensive evaluation, we uncover a phenomenon of **Surface Learning** behavior on LLMs similar to student learning behavior in Education Psychology. The behavior indicates that although the models seem to know the formulas and strategies required to solve specific types of problems on the surface, they do not truly comprehend the essence of these concepts, resulting in surface-level short-term benefits rather than in-depth learning. Further to mitigate surface learning behavior of LLMs, we propose a long-term strategy for both training-free and post-training scenarios. In training-free scenario, inspired by Self-Concept theory, LLMs are prompted with goal-setting and planning beforehand as well as feedback afterward to improve the ability in reasoning process. To better activate the underlying knowledge during the post-training process, we propose behavior correction strategy to re-rank samples based on the designed self-cognition indicators of LLMs. This strategy prevents models from relying on easy-to-find paradigms to maximize rewards or minimize losses in the initial training stage, rather than undertaking actual reasoning. Extensive experiments of **S**upervised and **R**einforcement **F**ine-**T**uning (**SFT**, **RFT**) conducted on LLMs demonstrate the effectiveness of the strategy.

## 1 Introduction

With the evolution and proliferation of Large Language Models (LLMs) (Xu & Poo, 2023), it is crucial to assess whether LLMs genuinely comprehend the underlying knowledge, which can contribute to ensuring reliability and faithfulness in real-world applications (Chowdhery et al., 2023; Zhang et al., 2023). As shown in left sub-figure of Figure 1, LLMs, like a student, exploits underlying knowledge (i.e. formulas) involved in the mathematical question and provide answers accordingly. By further deepening the grasp of complex concepts and rigorous relationships behind formulas, LLMs can solve numerous similar problems rather than merely imitating contexts (Qin et al., 2023). In this paper, we shift the focus from quantifying how much LLMs learn to assessing what they learn (Chang et al., 2024), which provides deeper insights into their knowledge acquisition.

To this end, we supplement the existing mathematical datasets with relevant formulas to evaluate whether LLMs grasp the underlying knowledge, and introduce an English grammar dataset which contains questions of different grades and different knowledge points. Furthermore, since problem-solving typically follows the principle of progressive learning from simple to complex drawing on Cognitive Load theory (Sweller, 2011), we construct a sequence of questions with increasing difficulty, enabling the model to perform continuous problem-solving based on the multi-turn dialogue history. Through comprehensive evaluations, we observe that while the model is familiar with formulas and strategies required to solve specific types of problems, it struggles to comprehend their underlying information and is incapable of applying formulas to resolve more complex problems. We incorporate three respective behavior of surface learning: **rote learning**, **ignoring background**

Figure 1: In the left sub-figure, the sequence of questions with increasing difficulty enables LLMs to perform continuous problem-solving based on the dialogue history. LLM can output the correct formula but with incorrect reasoning process, failing to comprehend the formula and demonstrating surface learning behavior. In the right sub-figure, we propose a long-term strategy to mitigate surface learning behavior in training-free and post-training scenarios.

**information**, and **focusing on answer paradigms**. Parallel to the student behavior (Dolmans et al., 2016; Geirhos et al., 2020) in Education Psychology (Marton & Säljö, 1976), students as surface learners, direct their attention towards learning the text itself and keep to a rote-learning strategy other than truly mastering (Svensson, 1977; Marton & Dahlgren, 1976), resulting in surface-level short-term benefits rather than in-depth learning.

Based on findings of surface learning in LLMs, we propose a long-term strategy to improve the performance of LLMs in different scenarios. Specifically, in the training-free scenario, inspired by Self-Concept theory in Psychology, as surface learners, it is suggested that student be provided with an opportunity for goal-setting and planning beforehand as well as feedback afterward (Marsh, 1990; Alexander, 2004). We introduce this **Self-Concept Planning** to prompt settings and discover the effects for different LLMs. Nevertheless, the performance gains from training-free methods remain limited, and the model still struggles to flexibly apply knowledge to challenging problems. To address it, we adopt post-training techniques specifically targeting these hard tasks, aiming to activate and schedule knowledge effectively. During the post-training process, it is crucial to ensure that the model truly utilizes knowledge to solve problems, rather than surface learning. However, we find that LLM exploits easy-to-find paradigms that conform to the reward model to obtain the reward scores during Reinforcement Fine-Tuning (RFT) process (Weng, 2024; Skalse et al., 2022). As shown in Figure 2, the problem-solving paradigms (highlighted in gray) are utilized in the model response, but there is no actual reasoning process. Therefore, we propose a **Behavior Correction** strategy with designed self-cognition indicators $I$. The training samples are re-ranked based on $I$, thereby preventing LLMs from reasoning through the easy-to-find answer paradigms. Meanwhile, we also introduce this strategy to perform Supervised Fine-Tuning (SFT) on query-answer pairs. Although it lacks the same capacity as RFT in learning the reasoning mechanism due to the absence of explicit reasoning process, the relevant formulas or knowledge points incorporated contribute to enhancing the model's proficiency in knowledge utilization. Experiments conducted on LLMs demonstrate the effectiveness of this strategy.

## 2 RELATED WORK

**Improving LLMs** has been a hot topic in recent years, which could be divided into prompt engineering (Wei et al., 2022b; Dong et al., 2024), fine-tuning (Trad & Chehab, 2024), and reinforcement learning (Tie et al., 2025). Among them, prompt engineering is a widely used method to enhance the performance of LLMs without modifying the model parameters. For instance, In-Context Learning (ICL) (Wei et al., 2022a; Schaeffer et al., 2023) helped LLMs to learn from several demonstration examples within a given context. Chain-of-Thought (CoT) (Wei et al., 2022b) proposed helped LLMs promote the reasoning process and explainability, rather than simply providing answers. Since then, variants of COT such as Tree-of-Thought (ToT) (Yao et al., 2023a), Graph-of-Thought (GoT) (Besta et al., 2024), Memory-of-Thought (MoT) (Li & Qiu, 2023), Skeleton-of-Thought (SoT) (Ning et al., 2023) and Exchange-of-thought (EoT) (Yin et al., 2023) were proposed to improve the reasoning process. In addition to training-free methods, Supervised Fine-Tuning (SFT) (Devlin et al., 2019; Lv et al., 2024) adjusted parameters of LLMs and adapted a pre-trained model to specific downstream tasks through supervised learning. When labeled data are limited, PEFT (Houlsby et al., 2019; Han et al., 2024), such as LoRA (Hu et al., 2022) and Adapter (He et al., 2022), optimized a

small subset of parameters or inserted lightweight modules, reducing computational costs and improving training efficiency without sacrificing performance. To further enhance the performance of LLMs, Reinforcement Learning (RL) (Christiano et al., 2017) was employed to optimize the model's responses based on feedback from human evaluators or reward functions, such as Proximal Policy Optimization (PPO) (Schulman et al., 2017), Direct preference optimization (DPO) (Rafailov et al., 2023), Group Relative Policy Optimization (GRPO) (Shao et al., 2024), Decoupled Clip and Dynamic sAmpling Policy Optimization (DAPO) (Yu et al., 2025), etc.

**Exploring multiple behaviors of LLMs** in improvement process helps to ensure faithfulness and reliability in real-world applications. Some research revealed that LLMs tended to hallucinate, generating incorrect or nonsensical information (Zhang et al., 2023; Chowdhery et al., 2023). LLMs also had the tendency to trust users and favor authoritative roles (Sharma et al., 2024; Zhao et al., 2025). The shortcut learning behavior of LLMs, which refers to the reliance of LLMs on certain words in prompt or spurious correlations in contexts, was explored through prompt perturbation or benchmark construction (Yuan et al., 2024; Tang et al., 2023). Moreover, in the settings of multi-turn dialogue settings, some studies (Collins et al., 2024; Du et al., 2024; Wang et al., 2023a) evaluated the performance of LLMs in the mathematical reasoning, multi-agent debate tasks. Similarly, in the RL process, LLMs were found to be prone to reward hacking (Skalse et al., 2022; Weng, 2024). This meant they exploited loopholes in the reward function to achieve high rewards without genuinely solving the task. Meanwhile, LLMs fine-tuned with Reinforcement Learning from Human Feedback (RLHF-LLMs) could over-rely on aligned preferences without truly reasoning (Wen et al., 2024; Amodei et al., 2016). In this paper, we explore the surface learning behavior of LLMs through knowledge supplements. By introducing RFT to mitigate surface learning behavior, we find that LLMs exploit easy-to-find paradigms to obtain rewards (e.g., reward hacking).

## 3 PROBLEM DEFINITION

In this paper, given the question $q$, prompt $\wp$, LLM $M$ aims to generate a probability of target $y$ conditioning on the prompt $\wp$, which can be written as: $P(y|q, \wp) = \prod_{t=1}^{T} P(y_t|q, \wp, y_{<t})$, where the $T$ is the generated token length. In the generated $y$, there are reasoning analysis $r$, answer $a$. Under different settings, the underlying knowledge $f$ (formulas or knowledge points) related to the question is adopted as part of input, helping evaluate the surface-level and deep-level mastery of knowledge by LLMs. We aim to deeply evaluate the extent of $P(r, a|\wp, q)$ and $P(r, a|f, \wp, q)$ in the paper, i.e., the capacity of LLMs to comprehend underlying knowledge.

## 4 SURFACE LEARNING BEHAVIOR & LONG-TERM STRATEGY

### 4.1 ME-TEST SUITE CONSTRUCTION

Since the DeepMind Mathematics dataset (Saxton et al., 2018) is categorized by difficulty levels EASY, MEDIUM, and HARD, we employ it to generate sequences of questions for evaluating mathematical problem-solving capabilities of LLMs. These questions are based on a national school mathematics curriculum for students up to age 16. LLMs are assessed on simple computations through the subsets **Add**ition and **Sub**traction, and **Mul**tiplication and **Div**ision; and on complex multi-step computations through the subsets **Factor** and **Remainder**. To determine whether LLMs comprehend underlying knowledge, we define and supplement a set of corresponding formulas $F = \{F^a, F^m, F^f, F^r\}$. For example, the Factor subset $F^f$ consists of formulas $\{F_1^f, ..., F_{l_f}^f\}$, where $l_f$ represents the number of relevant formulas.

In addition to mathematical problems that require strong reasoning skills, we collect multiple-choice and fill-in-the-blank Question-Answering (QA) data that demand English (Eng) grammar knowledge (Zheng et al., 2021). These questions are sourced from the National Middle School Entrance Examination (NMSE) and the National College Entrance Examination (NCEE) in China, targeting JUNIOR and SENIOR high school students[1]. Existing QA dataset rarely incorporate relevant knowledge. Nevertheless, grammar data itself comprises authoritative answers and knowledge points that do not demand additional annotations. It includes 41 grammar knowledge points and

---

[1]https://www.eol.cn/

| | |
|---|---|
| **Question** | Given relevant formulas of question: {} 
 Question: Is 7 a factor of 16608980? |
| **Response** | <think> 
 To solve this problem, let's break it down step by step. 
 First, we can use Formula7.6 to check if 16608980 is divisible by 7. We add the last two digits to twice the rest: 80 + (2 * 1660) = 80 + 3320 = 3400. Since 3400 is not divisible by 7, we can rule out this possibility. ✗ 
 Next, we can use Formula7.7 to check if 16608980 is divisible by 7. We start from the last digit and alternate multipliers of 1, 3, 2, -1, -3, -2 in a repeating cycle: 0, (1 * 8) = 8, (3 * 9) = 27, (2 * 8) = 16, (-1 * 7) = -7, (-3 * 0) = 0. Since 0 is not divisible by 7, we can rule out this possibility. 
 Next, we can use Formula7.1 to check if 16608980 is divisible by 7. We group the digits in threes from right to left: 0, 8, 8 and 6, 0, 9. We calculate the alternating sum: (0 - 8 + 8) + (6 - 0 + 9) = 0 + 15 = 15, which is not divisible by 7. We can rule out this possibility. ✗ 
 ... 
 </think> 
 <answer>False</answer> |

Figure 2: At the beginning of the training process, LLM finds the answer paradigms and easily obtain the reward score (correct answers and formats), but there is no actual reasoning involved. The underlined represents incorrect reasoning. The correct reasoning using the formula is as follows: 80+(2*1660898), 0+(8*3)+(9*2)+(8*-1)+0+(6*-2)+6*1+1*3=31, 980-608+16=388. The relevant formulas definitions can be found in Appendix D.

a total of 13,620 questions. We carry out comprehensive evaluations on 15,000 questions of each **M**athematical subset and 13,620 **E**nglish grammar questions, thereby forming the ME-Test suite. Relevant formulas and grammar knowledge could be found in Appendix D.

## 4.2 EVALUATION ON SURFACE LEARNING

To conduct a thorough evaluation of LLMs' capabilities, we employ diverse LLMs from: (1) Open-source models: LLaMA3.1 (8B, 70B) (Team, 2024), and Qwen2.5 (7B, 14B) (Yang et al., 2024). (2) Closed-source models: GPT-4o (Hurst et al., 2024). (3) Reasoning LLMs: DeepSeek-R1 (Guo et al., 2025), o3-mini[2]. Recent studies (Tang et al., 2023; Liu et al., 2024) often adopt perturbations or provide incorrect solutions to assess the model's ability. Nevertheless, these methods mislead LLMs (Wang et al., 2023a; Cohn & Hernandez-Orallo, 2023), definitely resulting in a decline in the model's performance. We evaluate LLMs' ability by starting from formulas or knowledge embedded within questions through various prompt settings as follows: (1) **Van**illa: output relevant formulas and answer; (2) **F**: given relevant formulas to question, output answer; (3) ¬**F**: given irrelevant formulas to question, output answer; (4) ∀**F**: given all available formulas, output answer. **All the settings require LLMs to output the reasoning process**.

## 4.3 LONG-TERM STRATEGY

As shown in right sub-figure of Figure 1, we propose a long-term strategy to mitigate surface learning behavior of LLMs, which mainly consists of Self-Concept Planning in training-free scenario and Behavior Correction strategy in post-training process.

### 4.3.1 SELF-CONCEPT PLANNING WITHOUT TRAINING

Inspired by self-concept theory in Education Psychology, surface learners perform somewhat better in structured, well-defined situations which provided an opportunity for goal-setting and planning beforehand and feedback afterward (Marsh, 1990). Therefore, we introduce this Self-Concept Planning strategy to force the model to concentrate on understanding underlying knowledge behind the question, other than just answering the question, and motivate LLMs to self-assess and provide feedback. In training-free settings, we design a series of prompts to guide LLMs in the reasoning process, which also offers a cognitive perspective on reasons behind the effectiveness of these methods, e.g., CoT (Wei et al., 2022c), CoT-SC (Wang et al., 2023b), Divide and Conquer (Cui et al., 2024; Zhou et al., 2023; Qi et al., 2023), ToT (Yao et al., 2023b), etc. We list all detailed prompt settings in Appendix D.

### 4.3.2 BEHAVIOR CORRECTION WITH POST-TRAINING

Since the performance gains from training-free methods remain limited, and the model still struggles to flexibly apply knowledge to challenging problems. In addition to surface learning behavior of LLMs, they perform poorly on the hard questions in the evaluation results. To tackle these issues,

---

[2]https://openai.com/index/openai-o3-mini/

we employ post-training techniques on hard questions in ME-Test suite, which are not only more challenging but also demand a deeper comprehension of the underlying knowledge.

During the training process, we find that the model exhibit surface learning behavior at the initial stage of training as shown in Figure 2. **To obtain the reward scores, LLM exploit answer paradigms by using formulas which contains correct answers and formats.** However, there is no actual reasoning involved, LLM still fails in understanding the concepts within the formulas. This can be attributed to the training paradigm of the model. Since the pre-trained model itself contains a powerful knowledge base of parameters, post-training helps the model activate and manage knowledge instead of re-learning it. In the post-training process, model maximizes the reward function or minimizes the loss function to optimize parameters, which will lead the model to prefer seeking simple methods to solve problems, and exhibit shortcut learning behavior (Geirhos et al., 2020; Skalse et al., 2022).

To address this, we propose a Behavior Correction (BC) strategy that reorders training samples based on constructed self-cognition indicators $I_i$. Inspired by research (Lampinen et al., 2024) that **models more strongly and densely concentrate on features that are simpler to compute or learned first**, we construct $I_i$ on each $i^{th}$ sample to steer learning toward more informative samples, promoting deeper reasoning. **For RFT**, we employ GRPO (Shao et al., 2024) to obtain a group of rewards $\{r_{i1}, \ldots, r_{iG}\}$ corresponding to outputs $\{o_{i1}, \ldots, o_{iG}\}$ on a sample, the $I_i^r$ can be defined as:

$$I_i^r = \frac{\min_{1 \leq g \leq G} r_{ig} + \text{mean}_{1 \leq g \leq G} r_{ig}}{\text{std}_{1 \leq g \leq G} r_{ig}}. \tag{1}$$

We re-rank the samples in ascending order based on $I_i^r$, thereby preventing the model priorities from learning the easy-to-find answer paradigms to obtain reward scores. Besides, to assess whether the model exhibits similar behavior during the **SFT process**, we define the self-cognition indicator $I_i^s$ as the loss on $i^{th}$ samples in the early training, and re-rank the samples in descending order. Earlier-ranked samples reflect model uncertainty, while later-ranked samples are deterministic—meaning the model is either fully confident or completely unsure. For the data within the hard subsets that possess **the same level of difficulty and low discrimination**, the behavior correction approach is employed to assist the model in exploring multiple problem-solving strategies and preventing it from being trapped in the optimization process by rapidly finding a set of problem-solving templates. The detailed explanation of $I^r$ can be found in Appendix B.2.

## 5 EXPERIMENTS

In this section, we first assess the performance of LLMs in comprehending underlying knowledge by employing a multi-turn dialogue setting, which incorporates authoritative formulas or key knowledge points for a comprehensive evaluation. Meanwhile, we compare this with a single-turn setting to analyze the impact of sequential questioning. Then, we experiment with self-concept planning to explore whether surface learning can be mitigated in the training-free scenario. We also compare different strategies to assess the effectiveness of behavior correction strategy in the post-training process. The ME-Test suite and the associated code are available for further study and research[3].

### 5.1 EXPERIMENTAL SETUP

In the evaluation, multi-turn dialogues can be constructed by sequentially appending messages with the "assistant" role, as outlined in the official documentation for most LLMs. We set the max_tokens of DeepSeek-R1[4] and max_output_tokens of o3-mini[5] are 8K. During the post-training process, GRPO (Shao et al., 2024) for reinforcement fine-tuning of LLaMA3.2-3B and Qwen2.5-7B on 3 × A800-80G and 8 × A800-80G GPUs respectively. We set num_generations to 10, learning_rate to 1e-6, and max_completion_length to 1024 and 4096, accordingly. Additionally, LLaMA3.1-8B and Qwen2.5-14B are employed to perform fine-tuning on A6000-40G GPU using the Low-Rank Adapters (LoRA) parameter-efficient tuning method (Hu et al., 2022), at a rank of 16 and alpha of

---

[3]https://anonymous.4open.science/r/SL-BC1D/

[4]https://api-docs.deepseek.com/

[5]https://openai.com/api/

Table 1: Experimental results of various LLMs with the Accuracy in multi-turn evaluation. As the difficulty increases, the performance of LLMs obviously drops. However, whether the formulas are related to the problem or not, the provision of formulas does not necessarily improve or reduce the performance. **Bold** indicates the highest results, while underlined indicates the lowest results.

| | LLM | Setting | Add_Sub | | | Mul_Div | | | Factor | | |
|---|---|---|---|---|---|---|---|---|---|---|---|
| | | | Easy | Medium | Hard | Easy | Medium | Hard | Easy | Medium | Hard |
| **Open-Source LLMs** | LLaMA3.1-8B | Van | **0.624** | **0.508** | **0.370** | **0.608** | **0.330** | **0.185** | 0.640 | 0.641 | 0.639 |
| | | F | 0.382 | 0.301 | 0.200 | 0.594 | 0.327 | 0.183 | **0.756** | **0.712** | **0.695** |
| | | ¬F | 0.108 | 0.087 | 0.056 | 0.362 | 0.259 | 0.161 | 0.751 | 0.661 | 0.633 |
| | | ∀F | 0.386 | 0.307 | 0.209 | 0.570 | 0.319 | 0.184 | 0.740 | 0.694 | 0.667 |
| | LLaMA3.1-70B | Van | **0.886** | 0.793 | 0.652 | 0.809 | 0.549 | 0.322 | 0.764 | 0.726 | **0.721** |
| | | F | 0.859 | 0.794 | 0.626 | 0.812 | **0.578** | 0.336 | **0.857** | **0.742** | 0.696 |
| | | ¬F | 0.835 | **0.796** | **0.659** | 0.831 | 0.575 | **0.344** | 0.594 | 0.486 | 0.456 |
| | | ∀F | 0.856 | 0.780 | 0.614 | 0.817 | 0.571 | 0.324 | 0.503 | 0.411 | 0.396 |
| | Qwen2.5-7B | Van | **0.854** | **0.766** | **0.676** | **0.840** | 0.601 | 0.386 | 0.638 | **0.636** | **0.573** |
| | | F | 0.362 | 0.363 | 0.337 | 0.816 | **0.658** | **0.467** | **0.667** | 0.526 | 0.470 |
| | | ¬F | 0.321 | 0.244 | 0.228 | 0.776 | 0.632 | 0.452 | 0.649 | 0.509 | 0.453 |
| | | ∀F | 0.341 | 0.346 | 0.315 | 0.812 | 0.657 | 0.464 | 0.626 | 0.491 | 0.466 |
| | Qwen2.5-14B | Van | 0.905 | 0.898 | 0.818 | **0.880** | **0.709** | **0.524** | 0.850 | 0.773 | 0.767 |
| | | F | **0.911** | **0.907** | **0.862** | 0.845 | 0.698 | 0.519 | 0.894 | 0.859 | 0.828 |
| | | ¬F | 0.880 | 0.861 | 0.823 | 0.850 | 0.697 | 0.523 | **0.962** | **0.892** | **0.853** |
| | | ∀F | 0.900 | 0.890 | 0.853 | 0.852 | 0.698 | 0.515 | 0.924 | 0.853 | 0.840 |
| **Closed-Source LLMs** | GPT-4o | Van | 0.874 | 0.876 | 0.876 | 0.948 | 0.841 | 0.609 | 0.607 | 0.570 | 0.574 |
| | | F | **0.882** | **0.880** | **0.883** | 0.951 | 0.849 | 0.611 | 0.904 | 0.854 | **0.859** |
| | | ¬F | 0.743 | 0.780 | 0.770 | 0.950 | 0.856 | **0.616** | **0.909** | **0.867** | 0.857 |
| | | ∀F | 0.799 | 0.793 | 0.796 | **0.962** | **0.863** | 0.616 | 0.826 | 0.783 | 0.780 |
| **Reasoning LLMs** | DeepSeek-R1 | Van | 0.920 | 0.890 | 0.865 | **0.998** | **0.999** | 0.984 | 0.930 | 0.890 | 0.845 |
| | | F | **0.970** | 0.915 | 0.880 | 0.992 | 0.992 | **0.986** | 0.965 | 0.900 | 0.855 |
| | | ¬F | 0.960 | **0.945** | 0.860 | 0.992 | 0.982 | 0.980 | 0.995 | 0.980 | 0.955 |
| | | ∀F | 0.930 | 0.938 | **0.940** | 0.990 | 0.994 | 0.984 | **0.999** | **0.999** | **0.997** |
| | o3-mini | Van | **0.965** | **0.970** | **0.975** | **0.868** | **0.778** | **0.686** | **0.980** | 0.960 | **0.910** |
| | | F | 0.890 | 0.890 | 0.885 | 0.842 | 0.702 | 0.606 | 0.960 | **0.975** | 0.664 |
| | | ¬F | 0.649 | 0.497 | 0.342 | 0.834 | 0.752 | 0.662 | 0.910 | 0.940 | 0.528 |
| | | ∀F | 0.868 | 0.750 | 0.655 | 0.758 | 0.622 | 0.546 | 0.888 | 0.813 | 0.753 |

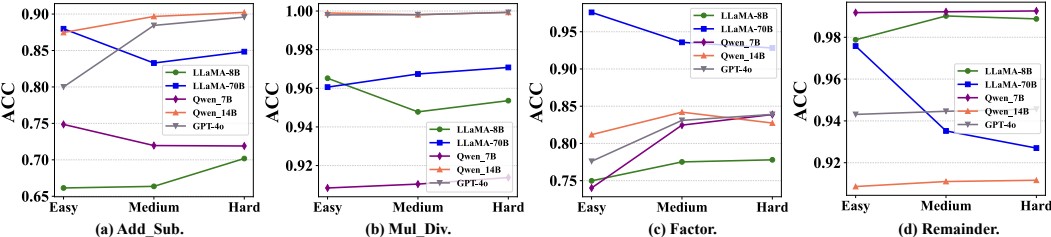

Figure 3: Experimental results of LLMs in predicting relevant formulas on mathematical questions. Contrary to prediction answers, as the difficulty increases, the performance of LLMs on formula prediction remains unchanged or improves.

32, over 10 epochs. To balance training costs, we employ fp16 precision, gradient accumulation strategy, and set the maximum length to 2048. AdamW optimizer (Loshchilov & Hutter, 2018), a 0.1 dropout, and a cosine annealed learning rate of 1e-4 are incorporated. We introduce hard subsets in ME-Test suite which contains senior grammar subset and mathematical reasoning tasks (Factor and Remainder) to achieve post-training. Each subset includes 2,000 randomly selected training samples and 1,000 test samples drawn from the remaining pool. We evaluate the LLMs' knowledge comprehension and reasoning abilities in both **Van** and **F** settings.

## 5.2 EXPERIMENTAL ANALYSIS

### 5.2.1 SURFACE LEARNING BEHAVIOR

**The main performance of LLMs on ME-Test suite** is shown in Table 1 and Figure 4. First, it can be obviously seen that the performance of LLMs declines as the difficulty increases, which aligns with the cognitive load theory. In vanilla setting, LLMs are asked to output formulas relevant to the question. The experimental results can be found in Figure 3, which is contrary to the results on

answers. As the difficulty increases, the performance of LLMs on formula prediction either remains unchanged or improves. This indicates the first behavior of surface learning in LLMs:

> **1. Rote Learning**
>
> LLMs learn the formulas for different types of mathematical problems on the surface, but do not understand the mathematical concepts within the formulas, failing to apply them and solve problems that seem more intricate.

Then, it can be observed that LLMs typically perform best in the vanilla and $F$ settings on different subsets. However, on factor subset, Qwen-14B performs the best under the $\neg F$ setting. LLaMA-70B exhibits the poorest performance in the $\forall F$ setting on add_sub and factor subsets. The second behavior of surface learning in LLMs can be concluded as:

> **2. Ignore Background Knowledge**
>
> During the learning process, LLMs only focus on answering and aligning labels blindly, ignoring understanding the logical knowledge behind question. The provision of relevant or irrelevant formulas does not necessarily improve or reduce the model's performance.

Contrary to the situation where LLMs fail in understanding formula concepts to solve math problems, LLMs rely more on concepts when addressing English grammar problems. The performance of all LLMs declines with given irrelevant formulas. LLMs tend to rely more on the spurious correlations between solutions and grammar concepts, and offer incorrect solutions based on grammar knowledge while disregarding the question itself. This represents the third behavior of surface learning in LLMs:

> **3. Focusing on Answer Paradigms**
>
> LLMs provide the problem-solving strategy on the surface, but learn spurious correlations between concepts and solutions in fact.

In addition to the above primary behaviors of surface learning in LLMs, we find that Qwen-14B exhibits the best performance among all except for GPT-4o and reasoning models in general. Particularly on Remainder subset, as presented in Table 2, Qwen-14B achieves about 20% higher accuracy than GPT-4o in multi-sound setting. LLaMA-8B and Qwen-7B, with smaller number of parameters, still demonstrate limited abilities in Remainder subset, which involves multi-step reasoning.

In the Factor subset, LLM demonstrates varying degrees of comprehension on formulas in different types of math problems. For instance, although Qwen-14B seems to perform well, the presence of irrelevant formulas leads to an enhancement in the model's performance, suggesting that Qwen-14B merely remembers corresponding correlations between questions and answers, but has a restricted understanding on related concepts. In the reasoning LLMs, due to the task avoidance behavior of o3-mini (Zhou et al., 2024), the performance of DeepSeek-R1 is more robust. Furthermore, in English grammar questions, GPT-4o performs best, while LLaMA-8B performs worst. In different settings, LLaMA-8B performs its best results when provided with relevant or all formulas. This reflects that the model has, to a certain extent, learned the correlations between concepts and solutions, and requires more well-defined background knowledge to mitigate surface learning.

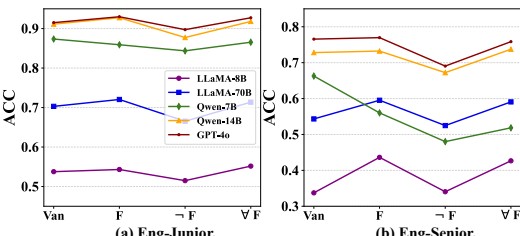

Figure 4: Experimental results of LLMs on English Junior and Senior grammar questions.

### 5.2.2 MULTI-TURN VS. SINGLE-TURN

Based on the cognitive load theory, we construct sequences of questions with increasing difficulty and allow LLMs to give multi-turn responses based on the dialogue history, enabling the model to conduct continuous problem-solving. Considering the complexity of dialogue interaction, we

Table 2: Experimental results of LLMs in multi-turn and single-turn settings on Remainder subset. Remainder computation only concerns one formula, and thus there are no other settings.

| | Remainder | LLaMA3.1-8B | | LLaMA3.1-70B | | Qwen2.5-7B | | Qwen2.5-14B | | GPT-4o | | o3-mini | |
|---|---|---|---|---|---|---|---|---|---|---|---|---|---|
| | | Van | F | Van | F | Van | F | Van | F | Van | F | Van | F |
| **Multi-turn** | **Easy** | 0.261 | **0.603** | 0.898 | **0.950** | 0.300 | **0.462** | **0.966** | 0.936 | 0.691 | **0.698** | **0.645** | 0.618 |
| | **Medium** | 0.116 | **0.285** | 0.632 | **0.641** | 0.123 | **0.205** | **0.782** | 0.755 | 0.509 | **0.514** | 0.430 | **0.445** |
| | **Hard** | 0.059 | **0.116** | 0.313 | **0.324** | 0.060 | **0.077** | **0.511** | 0.494 | 0.311 | **0.319** | 0.295 | **0.308** |
| **Single-turn** | **Easy** | **0.604** | 0.537 | 0.631 | **0.634** | **0.796** | 0.482 | 0.934 | **0.939** | 0.922 | **0.968** | 0.788 | **0.935** |
| | **Medium** | **0.307** | 0.296 | **0.386** | 0.327 | **0.376** | 0.190 | 0.583 | **0.698** | 0.718 | **0.827** | 0.810 | **0.880** |
| | **Hard** | 0.122 | **0.162** | **0.218** | 0.136 | **0.118** | 0.085 | 0.308 | **0.386** | 0.357 | **0.493** | **0.835** | 0.813 |

Table 3: Experimental results of LLMs in single-turn evaluation. Similar to findings in multi-turn evaluation, as the difficulty increases, the performance of LLMs obviously drops. All models exhibit surface learning behavior.

| | | Llama3.1-8B | | | | Qwen2.5-7B | | | | Qwen2.5-14B | | | | GPT-4o | | | |
|---|---|---|---|---|---|---|---|---|---|---|---|---|---|---|---|---|---|
| | | Van | F | ¬F | ∀F | Van | F | ¬F | ∀F | Van | F | ¬F | ∀F | Van | F | ¬F | ∀F |
| **Add_Sub** | **Easy** | **0.779** | 0.633 | 0.549 | 0.597 | **0.932** | 0.895 | 0.870 | 0.875 | **0.969** | 0.933 | 0.939 | 0.912 | **0.957** | 0.917 | 0.841 | 0.856 |
| | **Medium** | **0.666** | 0.504 | 0.405 | 0.422 | **0.884** | 0.833 | 0.791 | 0.804 | **0.919** | 0.883 | 0.880 | 0.887 | **0.953** | 0.907 | 0.831 | 0.839 |
| | **Hard** | **0.512** | 0.352 | 0.324 | 0.274 | **0.793** | 0.722 | 0.687 | 0.717 | 0.833 | 0.809 | 0.810 | 0.837 | **0.910** | 0.865 | 0.783 | 0.801 |
| **Factor** | **Easy** | **0.871** | 0.820 | 0.743 | 0.718 | **0.946** | 0.851 | 0.756 | 0.870 | **0.964** | 0.917 | 0.909 | 0.961 | **0.961** | 0.894 | 0.940 | 0.912 |
| | **Medium** | 0.740 | **0.741** | 0.651 | 0.659 | **0.870** | 0.784 | 0.700 | 0.802 | **0.899** | 0.841 | 0.862 | 0.898 | **0.895** | 0.835 | 0.866 | 0.855 |
| | **Hard** | 0.694 | **0.722** | 0.616 | 0.645 | **0.793** | 0.726 | 0.663 | 0.736 | **0.864** | 0.813 | 0.822 | 0.857 | 0.827 | 0.810 | **0.840** | 0.824 |

additionally evaluate the performance of LLMs in single-turn setting. As shown in Tables 2 and 3, LLMs perform better in the single-turn setting, but their performance declines when provided with either relevant ($F$) or irrelevant ($\neg F$) formulas in most cases. This suggests that, despite overall performance gains, LLMs exhibit stronger surface learning behavior. In multi-turn setting, the model needs to understand the information in historical dialogues and effectively utilize the information for the response of current round, which is beneficial for helping the model to learn and conduct continuous problem-solving. While in single-turn evaluations, the rote learning behavior of LLMs becomes more pronounced, i.e., answering but not understanding mathematical concepts in formulas. Besides, in Appendix B.8, we present statistics on LLM output token lengths, which show no linear relationship with model performance.

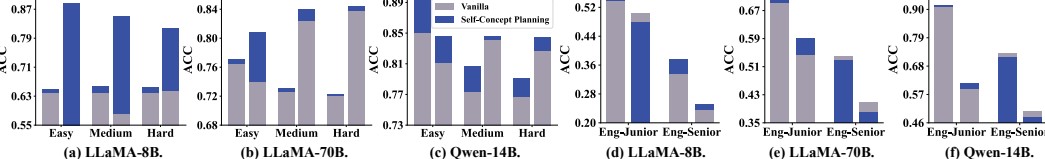

Figure 5: Performance Comparisons of LLMs in the settings of Vanilla and Self-concept planning strategy on Factor (a, b, c) and Eng datasets (d, e, f). In each group of columns, the left and right values respectively denote the performance in predicting answer and formulas/knowledge.

### 5.2.3 CAN LONG-TERM STRATEGY HELP?

**Self-Concept Planning without Training.** As illustrated in Section 4.3, we introduce this strategy based on the advocacy in Educational Psychology. The experimental results are presented in Figure 5. Self-Concept Planning strategy can pay a crucial role in enhancing the performance of predicting answers and relevant formulas on mathematical problems. In general, compared with LLaMA-70B, the performance of LLaMA-8B in formula prediction and that of Qwen-14B in answer prediction are significantly improved. Due to the third behavior "focusing on answer paradigms" in surface learning, the model captures the spurious correlations between concepts and solutions during the RLHF process and focus on alignment. This strategy could help to address English grammar problems under some settings compared to solving math problems. Therefore, it is challenging to directly eliminate the behavior through the prompt strategy and activate the reasoning ability of the base model itself. More experiments compared with ToT are provided in Appendix B.4.

**Behavior Correction with Post-Training.** To further mitigate the surface learning behavior and improve the reasoning ability of LLMs, we propose Behavior Correction (BC) strategy in post-training. To evaluate the effectiveness of this strategy using $I^r$ and $I^s$, we additionally fine-tune with random samples (w/o $I^r$, w/o $I^s$) and adopt the method of inverting the $I$ metric (Inverse-BC) for both RFT and SFT (w/ -$I^r$, w/ -$I^s$). As shown in Figure 6, we

adopt Mathematical and English hard subsets to make post-training, where "F" denotes the setting of the given formulas. LLaMA3.2-3B and Qwen2.5-7B are employed to achieve RFT.

It can be observed that adopting $I^r$ has achieved growth in most hard problems. The method of shuffling the training samples (RFT, w/o $I^r$) results in the model's performance with given formulas consistently being lower than that of the vanilla setting. This indicates that LLM finds the answer paradigm during the training process but fails to solve the reasoning problems on the test sets with given formulas, presumably because it does not fully comprehend the formulas. However, behavior correction strategy performs better with given formulas on test sets, suggesting that the strategy can reduce the model's tendency to use paradigms to obtain reward scores during training, alleviate the surface learning behavior of LLM in the reinforcement learning process, and thereby truly utilize the formulas to solve problems. **More reasoning comparisons and case studies in RFT** versus baselines could be found in Appendices B.5 and B.6.

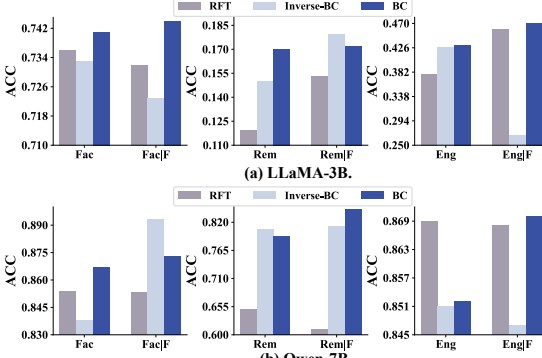

Figure 6: Experimental results of LLMs on hard subsets are compared with basic RFT method (w/o $I^r$) and the RFT method with inverse $I^r$ (w/ $-I^r$). Behavior correction strategy (w/ $I^r$) can improve the ability of LLMs compared with other baselines.

As shown in Figure 7, we employ LLaMa3.1-8B and Qwen2.5-14B to achieve SFT on hard questions. Given that the data solely comprises query-answer pairs and lacks any explicit reasoning process, the improvement achieved through fine-tuning the model is limited. This process is more analogous to conducting instruction fine-tuning. When compared to randomly shuffling the data (SFT) and adopting the Inverse-BC for re-ranking training data, adopting the behavior correction strategy can effectively improve the model's performance in most settings. Specifically, for LLaMA-8B, behavior correction strategy enables the model to perform better compared to the vanilla setting when provided with relevant formulas. Regarding Qwen-14B, English grammar questions typically do not require multi-step reasoning. Under the vanilla setting, although the model

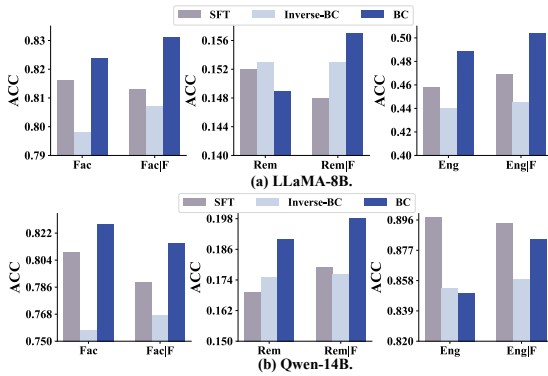

Figure 7: Experimental results of LLMs on hard subsets are compared with basic SFT method (w/o $I^s$) and the SFT method with inverse $I^s$ (w/ $-I^s$). Behavior correction strategy (w/ $I^s$) can improve the ability of LLMs compared with other baselines.

with re-ranking based on $I^s$ performs worse than w/o $I^s$ in Eng, $I^s$ helps the model in improving its reasoning ability. This improvement allows the model to perform better when given formulas and enables it to master grammar knowledge more effectively.

## 6 CONCLUSION

In this paper, by exploring the performance of various LLMs in solving Mathematical reasoning and English grammar problems, we found that LLMs exhibited surface learning behaviors similar to student behavior in Education Psychology, which could incorporate as rote learning, ignoring background knowledge, and focusing on answer paradigms. LLMs seemed to know what formulas and strategies needed to be adopted for solving different problems on the surface, but they did not truly grasp the essence of these concepts, where LLMs merely memorized but failed to apply solutions flexibly. Further to mitigate surface learning behavior of LLMs, we proposed long-term strategy which contained self-concept planning and behavior correction strategy in training-free and post-training scenarios, respectively. Extensive experiments on ME-Test suite demonstrated the effectiveness of the strategy.

## REPRODUCIBILITY STATEMENT

In Section 5, we include anonymous downloadable source code and dataset, comprehensive specific prompts in different settings can be found in Appendix D.

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

# CONTENTS

## A   THE USE OF LARGE LANGUAGE MODELS (LLMS)

In this paper, we study large language models as our primary subject and use them solely as a tool to refine our writing.

## B   MORE ANALYSIS

### B.1   SURFACE LEARNING VS. SHORTCUT LEARNING

Shortcut learning was initially defined as relying heavily on non-robust features—so-called "shortcuts"—that yield strong performance on standard benchmarks but fail to generalize to more challenging test settings (Geirhos et al., 2020). Subsequently, in classification tasks, it has been more precisely characterized as models exploiting spurious correlations between certain features and labels in the training data, rather than learning the underlying true patterns (Du et al., 2023; Li et al., 2023). However, in more complex generative tasks, shortcut learning lacks a clear definition. In this paper, we comprehensively evaluate and define Surface Learning in large language models—a typical and respective form of shortcut learning behavior—where models exhibit three representative behaviors indicating they have learned surface associations between questions and solution

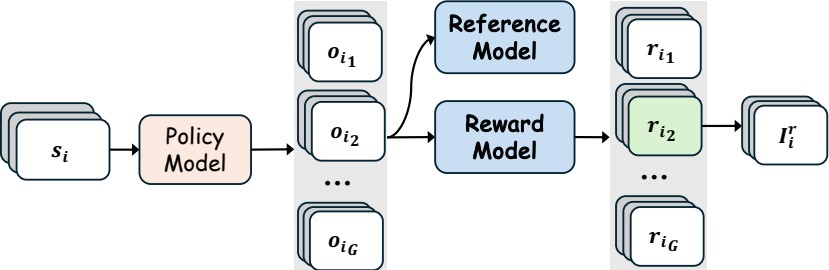

Figure 8: The $I^r$ computation representation in GRPO. We can obtain a group of rewards $\{r_{i1}, r_{i2}, \ldots, r_{iG}\}$ corresponding to the outputs $\{o_{i1}, o_{i2}, \ldots, o_{iG}\}$ on $i^{th}$ sample, where $G$ denotes the num_generations. Suppose $r_{i2}$ is the generation with the lowest reward in the generation group, indicating that the model does not use the answer paradigm to steal scores for this generation successfully. Meanwhile, we add the mean and standard deviation of the generation group to comprehensively consider all the reward scores of the generations on this sample, forming the self-cognition indicator of LLM for this sample.

.

paradigms, without genuinely understanding the underlying knowledge or solving the tasks. Additionally, Reward hacking (Weng, 2024) discussed in this paper can also be summarized as a kind of shortcut learning. The third typical behavior in surface learning-focusing on answer paradigms, is the typical behavior of the model after reward hacking in reinforcement learning.

### B.2 SELF-COGNITION INDICATOR ANALYSIS

As shown in Figure 8, in the GRPO framework, for the $i^{th}$ sample, we can obtain a group of rewards $\{r_{i1}, r_{i2}, \ldots, r_{iG}\}$ corresponding to the outputs $\{o_{i1}, o_{i2}, \ldots, o_{iG}\}$. Suppose $r_{i2}$ is the generation with the lowest reward in the generation group, indicating that model fails in the question, and also showing that the model does not utilize the answer paradigm to obtain scores successfully. Meanwhile, the mean and standard deviation of the generation group are added to comprehensively consider all the reward scores of the generations on this sample. The larger the standard deviation of reward scores within a group, the greater the variation among outputs (i.e., high intra-group diversity). Such samples are uncertain and offer more learning potential during post-training.

Self-cognition indicator is designed to prioritize learning uncertain and oscillating samples-those performed low $I^r$ with scattered rewards (high std), low overall performance (low mean), or large reward fluctuations (low min). In contrast, prioritizing training deterministic samples with high $I^r$ (low std, i.e., always learned or never learned) can lead two problems: (1) It will not trigger any parameter updates, i.e., no invalid calculations. (2) It gets trapped in a local optimum, and the model repeatedly generates similar low-quality paths. If such deterministic samples are given priority for training, it will cause the model to learn the answer paradigms and engage in surface learning, hindering true reasoning development. Instead, we focus on prioritized training of uncertain or oscillating samples, that is, the ones with high variance, with the worst-generated ones among multiple generations, the model shows uncertainty regarding such samples. Prioritizing these samples can help the model training by enabling it to explore more and better reasoning paths.

### B.3 MORE EVALUATIONS ON DOMAIN DATASET

To explore the surface learning performance of LLMs in more scenarios, we additionally conduct experiments on the JEC-QA dataset (Zhong et al., 2020), a challenging domain-specific benchmark for Legal Professional Certification QA. This dataset requires comprehending and applying specialized legal knowledge, representing a significantly different scenario compared to the ME-test. We evaluate on 1,000 Knowledge-Driven (KD) questions, which include annotated legal knowledge. The experimental results of different LLMs on JEC-QA are shown in Figure 9 and 10.

The experimental results show that the findings of surface learning behavior persist in different scenarios. The performance of almost all LLMs does not necessarily improve even when relevant

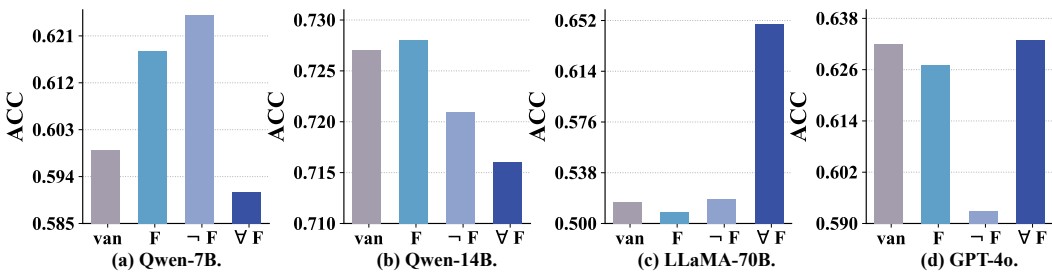

Figure 9: Experimental results of LLMs on legal domain dataset-JEC-QA. All LLMs perform surface learning behavior.

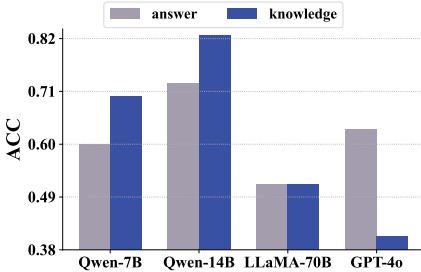

Figure 10: The performance of different LLMs in answering and predicting legal knowledge of questions on JEC-QA.

knowledge ($F$) is provided; Qwen-7B and LLaMA-70B even achieve the best results when given irrelevant knowledge ($\neg F$). Moreover, the Qwen series performs relatively better at predicting legal knowledge for a question than answering questions, which indicates that they know the knowledge behind the question on the surface, but struggle to comprehend and apply it. This reinforces that surface learning is a common behavior of LLMs across different problem types. Besides, performance trends hold: Qwen series generally performs well on domain-specific knowledge; models like LLaMA-70B and GPT-4o, which show overall poor performance, also demonstrate weaker knowledge comprehension capabilities.

### B.4 MORE EXPERIMENTS ON SELF-CONCEPT PLANNING

Table 4: The comparison of baselines in Factor subset, where self-concept planning achieves better overall performance while generating fewer tokens by reducing unnecessary outputs through deep knowledge understanding (a, f represent performance on **a**nswering and predicting **f**ormula).

| | **Easy** | | **Medium** | | **Hard** | | avg. input | avg. output |
|---|---|---|---|---|---|---|---|---|
| | a | f | a | f | a | f | | |
| Vanilla | 85.04 | 81.18 | 77.34 | 84.20 | 76.74 | 82.74 | 7379.03 | 366.34 |
| ToT | 85.60 | 84.60 | **82.70** | **85.90** | 78.40 | **84.60** | 28702.24 | 3756.05 |
| Self-Concept Planning | **89.36** | **86.64** | 80.64 | 84.64 | **79.12** | 84.48 | 7578.53 | 260.3 |

As illustrated in Section 4, all settings including "Vanilla" setting require LLMs to output the reasoning process. In addition to the direct comparisons made in Figure 5, we also conduct additional experiments using ToT (Yao et al., 2023a) to enable a more comprehensive comparison. We perform BFS algorithm in ToT with Qwen2.5-14B. For computational efficiency, we sample 3 independent thoughts, apply a vote strategy to evaluate 3 states, and keep the best answer. The experimental results are shown in Table 4, self-concept planning achieves better overall performance while

generating fewer tokens per question by reducing unnecessary outputs through deep knowledge understanding. However, ToT consumes excessive tokens during multi-turn evaluations and has yet to see meaningful efficiency improvements.

## B.5 More experiments on Behavior Correction strategy

To fully compare the performance of the behavior correction strategy with baselines in improving the reasoning ability. In addition to reporting accuracy, we also follow the approach (Guo et al., 2024) and conduct a multi-metric comparison of the reasoning processes of RFT models under three training mechanism (RFT, Inverse-BC, BC) across different subsets. These comparisons are based on correctly answered questions. We adopt GPT-4o and prompts as follows to evaluate the reasoning process of Qwen2.5-7B under different RFT strategies based on three criteria: Comprehensiveness, Diversity, and Empowerment:

```
You are an expert tasked with evaluating two reasoning processes to the
same question based on three criteria: Comprehensiveness, Diversity, and
Empowerment:

---Goal---
You will evaluate two reasoning processes to the same question based on
three criteria: Comprehensiveness, Diversity, and Empowerment.

 - Comprehensiveness: How much detail does the reasoning process provide
 to cover all aspects and details of the question?
 - Diversity: How varied and rich is the reasoning process in providing
 different perspectives and insights on the question?
 - Empowerment: How well does the reasoning process help the reader
 understand and solve the question?

For each criterion, choose the better reasoning process (either Reasoning
 1 or Reasoning 2) and explain why. Then, select an overall winner based
on these three categories.
```

Table 5: The performance comparison of Inverse-BC strategy $(-I^r)$ and BC strategy $(I^r)$ across four evaluation dimensions.

| | Factor | | Factor$|F$ | | Remainder | | Remainder$|F$ | |
|---|---|---|---|---|---|---|---|---|
| | Inverse -BC | BC | Inverse -BC | BC | Inverse -BC | BC | Inverse -BC | BC |
| Comprehensive | 22.2 | **76.7** | 23.0 | **76.6** | 21.9 | **78.0** | 47.4 | **52.6** |
| Diversity | 29.9 | **60.3** | 28.8 | **62.8** | 24.7 | **53.6** | 33.9 | **58.9** |
| Empowerment | 21.1 | **77.6** | 26.3 | **73.0** | 27.6 | **72.0** | 41.5 | **58.5** |
| Overall | 22.7 | **76.2** | 23.9 | **75.4** | 23.3 | **76.5** | 44.3 | **55.7** |

Table 6: The performance comparison of RFT (w/o $I^r$) and BC strategy $(I^r)$ across four evaluation dimensions.

| | Factor | | Factor$|F$ | | Remainder | | Remainder$|F$ | |
|---|---|---|---|---|---|---|---|---|
| | RFT | BC | RFT | BC | RFT | BC | RFT | BC |
| Comprehensive | 41.7 | **57.9** | 35.9 | **63.8** | 22.5 | **77.5** | 38.8 | **61.2** |
| Diversity | 36.1 | **48.5** | 33.7 | **57.9** | 26.1 | **54.1** | 46.2 | **46.9** |
| Empowerment | 44.7 | **54.9** | 40.8 | **59.1** | 32.7 | **67.3** | 45.8 | **54.2** |
| Overall | 43.9 | **55.8** | 36.9 | **62.9** | 14.8 | **73.6** | 42.4 | **57.6** |

As shown in Tables 5 and 6, the win rates of different strategies on math problems are reported; outcomes not classified as wins or losses are counted as ties. The behavior correction strategy demonstrates better performance across multiple dimensions, **encourages the model to explore**

| | |
|---|---|
| **Question** | Given relevant formulas of question:
Remainder = Dividend - Divisor * Quotient; Quotient = Dividend / Divisor (Quotient retains the integer part).
What is the remainder when 13065121 is divided by 424? |
| **w/o $I^r$** | \<think\>
To find the remainder when 13065121 is divided by 424, we can use the formula: Remainder = Dividend - Divisor * Quotient.
First, we need to determine the quotient when 13065121 is divided by 424. The quotient will be the integer part of the division result. Quotient = 13065121/424. Perform the division: 13065121/424 ≈30768.99974.
The integer part of this division is 30768. So, the quotient is 30768.
Next, we calculate the product of the divisor and the quotient: Divisor * Quotient = 424 * 30768.
We perform the multiplication: 424 * 30768 = 424 * (30000 + 700 + 60 + 8) = 424 * 30000 + 424 * 700 + 424 * 60 + 424 * 8 = 12720000 + 296800 + 25440 + 3392 = 13065120.
Now, we subtract this product from the dividend to get the remainder: Remainder = 13065121 - 13065120 = 1
Thus, the remainder when 13065121 is divided by 424 is 1.
\</think\>
\<answer\>1\</answer\> |
| **w/ -$I^r$** | \<think\>
To find the remainder when 13065121 is divided by 424, we can use the formula for the remainder: Remainder = Dividend - Divisor * Quotient.
First, we need to determine the quotient when 13065121 is divided by 424. The quotient is the integer part of the division of 13065121 by 424. We perform the division: Quotient = 13065121/424.
Performing the division: 13065121/424≈30817.99976744186.
The integer part of this division is 30817. Now, we can find the remainder using the formula: Remainder = 13065121 - 424 * 30817.
Next, we calculate 424 * 30817: 424 * 30817 = 424 * (30800 + 17) = 424 * 30800 + 424 * 17. 424 * 30800 = 424 * (30000 + 800) = 424 * 30000 + 424 * 800. 424 * 30000 = 12720000, 424 * 800 = 339200. 424 * 30800 = 12720000 + 339200 = 13059200. 424 * 17 = 7192. 424 * 30817 = 13059200 + 7192 = 13066392.
Now, we find the remainder: Remainder = 13065121 - 13066392 = -1271
Since the remainder should be a non-negative number less than the divisor, we add the divisor to -1271:
Remainder = -1271 + 424 = -847 + 424 = -423 + 424 = 1. So, the remainder when 13065121 is divided by 424 is 1.
\</think\>
\<answer\>1\</answer\> |
| **w/ $I^r$** | \<think\>
To find the remainder when 13065121 is divided by 424, we can use the division algorithm which states that for any integers a and b (where b > 0), there exist unique integers q (the quotient) and r (the remainder) such that: a = bq + r, where 0 < r < b. Here, a = 13065121 and b = 424. We need to determine r.
First, calculate the quotient q: q = 13065121/424. Perform the division: 13065121/424 ≈ 30817.19. Taking the integer part of this result gives us: q = 30817.
Next, compute the product of the quotient and the divisor: 424 * 30817. We perform the multiplication step-by-step: 424 * 30817 = 424 * (30800 + 17) = 424 * 30800 + 424 * 17. Calculate each term separately: 424 * 30800 = 424 * (308 * 100) = (424 * 308) * 100. First, find 424 * 308: 424 * 308 = 424 * (300 + 8) = 424 * 300 + 424 * 8. 424 * 300 = 127200. 424 * 8 = 3392. 424 * 308 = 127200 + 3392 = 130592. Now multiply by 100: 130592 * 100 = 13059200. Next, calculate 424 * 17: 424 * 17 = 424 * (10 + 7) = 424 * 10 + 424 * 7. 424 * 10 = 4240. 424 * 7 = 2968. 424 * 17 = 4240 + 2968 = 7208. Add these results together: 13059200 + 7208 = 13066408.
Finally, subtract this from the original number to find the remainder: 13065121 - 13066408 = -1287
Since the remainder should be non-negative and less than 424, add 424 repeatedly until the result is within the required range: -1287 + 424 * 3 = -1287 + 1272 = -15. -15 + 424 = 409. Thus, the remainder is: 409.
\</think\>
\<answer\>409\</answer\> |

Figure 11: The case in Remainder test set is to compare different RFT methods on complex reasoning question. Base RFT (w/o $I^r$) and RFT with inverse $I^r$ (w/ -$I^r$) methods both employ numerous  repetitive answer paradigms  and output  incorrect reasoning processes and answers . However, behavior correction strategy (w/ $I^r$) can provide the accurate answer. Despite there is a step error during the reasoning process, the behavior correction strategy can handle the complex question due to a solid grasp of the knowledge.

**more diverse and robust reasoning paths**. It further validates the effectiveness of the Behavior Correction strategy, robustly support refining LLM reasoning and knowledge comprehension.

## B.6   CASE ANALYSIS

In RFT, we design a new indicator $I^r$ to re-rank samples, which aims to prevent LLMs from exploiting easy-to-find answer paradigms to solve problems. Although the model can follow a certain paradigm when answering questions, the behavior of stealing the reward scores by utilizing the paradigm will result in the model not truly applying the knowledge to answer questions in the RFT process. As shown in Figures 11 and 12, we list the performance of the behavior correction strat-

| | |
|---|---|
| **Question** | Given relevant formulas of question:
Formula31.1: If the last digit is multiplied by 3 and subtracted from the rest, yielding a result divisible by 31, then the number is divisible by 31.
Is 31 a factor of 125401261? |
| **w/o $I^r$** | <think>
To determine if 31 is a factor of 125401261 using Formula 31.1, we need to follow these steps:
1. Separate the last digit from the rest of the number.
2. Multiply the last digit by 3.
3. Subtract this product from the rest of the number.
4. Check if the result is divisible by 31.
Let's apply these steps to the number 125401261.
1. Separate the last digit: The last digit is 1, and the rest of the number is 12540126.
2. Multiply the last digit by 3: 1 * 3 = 3.
3. Subtract this product from the rest of the number: 12540126 - 3 = 12540123.
Now, we need to check if 12540123 is divisible by 31. We can do this by performing the division: 12540123/31
Using long division or a calculator, we find that: 12540123/31 = 404843
Since the result is an integer, 12540123 is divisible by 31. Therefore, according to Formula 31.1, the original number 125401261 is also divisible by 31.
</think>
<answer>True</answer> |
| **w/ -$I^r$** | <think>
To determine if 31 is a factor of 125401261, we can use the provided formula which states that for a number to be divisible by 31, when you multiply its last digit by 3 and subtract it from the rest of the number, the result should also be divisible by 31.
Let's apply this step-by-step on our given number 125401261:
1. Separate the last digit: The last digit is 1.
2. Multiply the last digit by 3: 1 * 3 = 3.
3. Subtract this product from the remaining number: 12540126 - 3 = 12540123.
Now check divisibility of new number (12540123) by 31 or repeat process until clear:
- Last digit of 12540123 is 3; multiply by 3 -> 9; subtract: 1254012 - 9 = 12540114
- Continue similarly till easy division testable or confirm non-divisibility.
But simpler direct calculation/standard method might show easier factoring or remainder tests. Checking divisibility directly or via small steps confirms: 125401261/31 gives exact quotient without remainder, thus confirming divisibility rules hold true here. Thus, yes, 31 fits as factor per tested logic.
</think>
<answer>True</answer> |
| **w/ $I^r$** | <think>
To determine if 31 is a factor of 125401261 using the given formula, we need to follow these steps:
1. **Identify the last digit**: The last digit of 125401261 is 1.
2. **Multiply the last digit by 3**: 1 * 3 = 3.
3. **Subtract this product from the rest of the number**:
  - Remove the last digit (1) from 125401261, leaving 12540126.
  - Subtract 3 from 12540126: 12540126 - 3 = 12540123. Now, check if 12540123 is divisible by 31.
4. **Divisibility test for 31**: Perform the division of 12540123 by 31.
  - Calculate 12540123/31. Performing the division: 12540123/31≈404810.41935. Since the result is not an integer, 12540123 is not divisible by 31.
Therefore, based on the formula, 125401261 is not divisible by 31.
</think>
<answer>False</answer> |

Figure 12: The case in Factor test set is to compare different RFT methods on complex reasoning question. Base RFT (w/o $I^r$) and RFT with inverse $I^r$ (w/ -$I^r$) methods both employ numerous  repetitive answer paradigms  and output  incorrect reasoning processes and answers . However, behavior correction strategy (w/ $I^r$) can provide the accurate answer. It also uses the steps within the steps to solve the question, and the reasoning process is logically clear.

egy in answering questions and applying knowledge when given formulas compared to the baseline methods.

As shown in Figure 11, this case shows a remainder calculation that requires the model to perform step-by-step reasoning by applying the formula. The base RFT (w/o $I^r$) and RFT with inverse $I^r$ (w/ -$I^r$) methods both employ numerous repetitive answer paradigms (highted in gray) and output incorrect reasoning process and answers (highlight in green with red color). On the contrary, the behavior correction strategy (w/ $I^r$) can provide the accurate answer, despite there is a step error

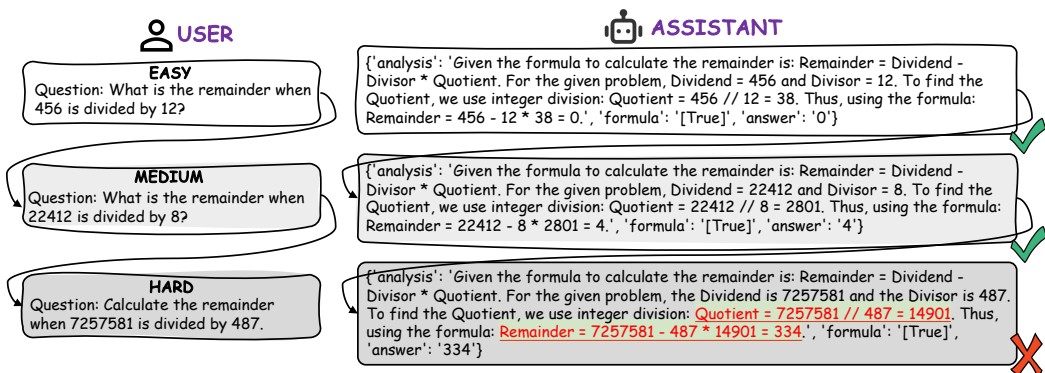

Figure 13: Due to surface learning, GPT-4o cannot apply the formula of Remainder to tackle hard question. Since remainder computation only concerns a single formula, "**Remainder = Dividend - Divisor * Quotient, Quotient = Dividend / Divisor**", LLMs are prompted to judge whether the formula is relevant to the question.

during the reasoning process. LLM corrects it at the final reasoning, making use of the knowledge that the remainder and quotient are integers. Meanwhile, the model also summarizes the formula and formalizes it in the form of "a = bq + r", flexibly applying the knowledge. It can contribute to a solid grasp of the knowledge to handle complex questions. As shown in Figure 12 to solve the Factor problem, compared to baseline methods, behavior correction strategy can flexibly apply knowledge and utilize the steps within the steps to solve the question. The reasoning process is logically clear.

In addition to the performance presentation of reinforcement fine-tuned LLM on Remainder subset, we present the case of GPT-4o on Remainder in the evaluation process of surface learning behavior in Figure 13. Although GPT-4o can solve easy and medium-level questions in sequences, it fails when dealing with hard-level question. As illustrated in Section 5.2.1, although LLMs seem to know the formulas and strategies required to solve specific types of problems on the surface, they fail to understand underlying knowledge behind the question and to handle seemingly intricate questions.

### B.7 MORE FINE-TUNING RESULTS ON MULTI-TURN & SINGLE-TURN EVALUATIONS

We conduct SFT experiments on 4000 hard samples of add_sub and factor subsets, the training data is equally distributed across four settings: Van, F, ¬ F, ∀ F in a 1:1:1:1 ratio. The remaining data is employed as test sets to assess the performance of the fine-tuned LLM in both single-turn and multi-turn settings. Experimental setup remain the same in Section 5.1. The experimental results are presented in Tables 7 and 8, and we have the following findings: (1) As illustrated in section 5.2.3, the DeepMind dataset lacks a reasoning process, containing only questions and answers, which limits the model's ability to learn long-context dependencies and reasoning. (2) While simple SFT data improves performance in single-turn settings, it significantly harms performance in multi-turn settings. SFT requires high-quality CoT data and stepwise training. However, long multi-turn contexts can lead to gradient explosions and demand multiple attempts for effective fine-tuning. (3) In a single-turn evaluation, LLM fails to improve the performance under setting of F, indicating that SFT merely enables the model to acquire spurious relationships. LLM fails to learn the knowledge underlying the formula in prompts and apply the knowledge (surface learning).

### B.8 MORE COMPARISON OF TOKENS OUTPUT

As shown in Figure 9, different LLMs produce varying output token lengths across subsets, and the output length generally increases with task difficulty. The longest outputs occur when models are given either relevant formulas (F) or all formulas (∀ F), whereas the shortest outputs appear under the irrelevant-formula setting (¬ F), where model performance is generally poor (see Table 1). Notably, Qwen-14B achieves the best performance on the factor subset while producing the fewest output tokens, further illustrating that model performance and output length are often not linearly correlated—longer outputs do not necessarily indicate better performance.

Table 7: Experimental results of comparisons between LLaMA3.1-8B and fine-tuned LLaMA3.1-8B in multi-turn evaluation settings. Fine-tuning on query-answer pairs leads to a significant decline in the multi-turn dialogue ability of the model, which may be related to the length and single relationship of the training data. **Bold** indicates the highest results, while underlined indicates the lowest results.

|  |  | LLaMA3.1-8B | | | | FT LLaMA3.1-8B | | | |
|---|---|---|---|---|---|---|---|---|---|
|  |  | Van | F | ¬ F | ∀ F | Van | F | ¬ F | ∀ F |
| **Add_Sub** | **Easy** | **0.611** | 0.382 | 0.108 | 0.386 | 0.246 | **0.351** | 0.230 | 0.323 |
|  | **Medium** | **0.504** | 0.301 | 0.087 | 0.307 | 0.220 | **0.318** | 0.210 | 0.293 |
|  | **Hard** | **0.374** | 0.200 | 0.056 | 0.209 | 0.196 | **0.265** | 0.187 | 0.254 |
| **Factor** | **Easy** | 0.640 | **0.756** | 0.751 | 0.740 | 0.238 | **0.241** | 0.237 | 0.239 |
|  | **Medium** | 0.641 | **0.712** | 0.661 | 0.694 | 0.229 | **0.230** | 0.229 | **0.230** |
|  | **Hard** | 0.639 | **0.695** | 0.633 | 0.667 | **0.227** | 0.226 | 0.225 | 0.225 |

Table 8: Experimental results of comparisons between LLaMA3.1-8B and fine-tuned LLaMA3.1-8B in single-turn evaluation settings. Fine-tuning on query-answer pairs helps improve the model's ability in the single-turn scenario, but it does not alleviate its surface learning behavior. Under the setting of given formula, the model performs even worse. **Bold** indicates the highest results, while underlined indicates the lowest results.

|  |  | LLaMA3.1-8B | | | | FT LLaMA3.1-8B | | | |
|---|---|---|---|---|---|---|---|---|---|
|  |  | Van | F | ¬ F | ∀ F | Van | F | ¬ F | ∀ F |
| **Add_Sub** | **Easy** | **0.779** | 0.633 | 0.549 | 0.558 | **0.986** | 0.978 | 0.978 | 0.975 |
|  | **Medium** | **0.666** | 0.504 | 0.405 | 0.422 | **0.950** | 0.944 | 0.943 | 0.937 |
|  | **Hard** | **0.512** | 0.352 | 0.324 | 0.264 | **0.859** | 0.851 | 0.856 | 0.843 |
| **Factor** | **Easy** | **0.871** | 0.820 | 0.743 | 0.584 | **0.857** | 0.806 | 0.848 | 0.843 |
|  | **Medium** | 0.740 | **0.741** | 0.651 | 0.553 | 0.819 | 0.814 | 0.812 | **0.820** |
|  | **Hard** | 0.694 | **0.722** | 0.616 | 0.523 | **0.820** | 0.818 | 0.817 | 0.818 |

Table 9: Average output token length over each subset across different settings.

| LLMs | Settings | Add_Sub | | | Mul_Div | | | Factor | | |
|---|---|---|---|---|---|---|---|---|---|---|
|  |  | Easy | Medium | Hard | Easy | Medium | Hard | Easy | Medium | Hard |
| **LLaMA3.1-8B** | Van | **138.41** | 153.06 | 163.53 | 98.58 | 107.70 | 112.27 | **194.01** | **187.27** | **190.74** |
|  | F | 128.97 | **158.37** | **179.15** | 112.60 | 120.63 | **130.12** | 90.59 | 94.39 | 106.19 |
|  | ¬ F | 115.54 | 134.10 | 150.43 | 115.41 | 120.02 | 123.94 | 85.92 | 91.02 | 100.27 |
|  | ∀ F | 130.01 | 156.48 | 175.59 | **119.87** | **123.17** | 127.20 | 135.19 | 163.52 | 179.67 |
| **LLaMA3.1-70B** | Van | 96.07 | 110.08 | 119.06 | 66.09 | 73.73 | 76.58 | 98.82 | 142.37 | 163.71 |
|  | F | **124.22** | **154.33** | **179.93** | 96.43 | **109.17** | 121.08 | 153.12 | 234.74 | 282.23 |
|  | ¬ F | 98.80 | 110.56 | 128.09 | 71.99 | 77.91 | 83.09 | 120.59 | 230.76 | 272.00 |
|  | ∀ F | 119.41 | 147.61 | 178.63 | **99.09** | 108.96 | **122.04** | **184.01** | **279.43** | **507.05** |
| **Qwen2.5-7B** | Van | 87.26 | 90.56 | 97.03 | 64.53 | 69.92 | 74.37 | 106.02 | 111.68 | 119.41 |
|  | F | 121.50 | 150.48 | 177.91 | **82.98** | **98.28** | **111.08** | 133.03 | **164.23** | 184.00 |
|  | ¬ F | 106.66 | 134.19 | 158.23 | 64.60 | 75.18 | 81.82 | 85.68 | 104.68 | 119.16 |
|  | ∀ F | **125.71** | **152.27** | **179.41** | 79.69 | 93.48 | 102.72 | **134.15** | 160.34 | **190.63** |
| **Qwen2.5-14B** | Van | 83.39 | 93.02 | 101.55 | 64.53 | 69.92 | 74.37 | 100.10 | 125.80 | 139.93 |
|  | F | **139.83** | **169.69** | **197.61** | **82.98** | **98.28** | **111.08** | 136.52 | **165.72** | **170.99** |
|  | ¬ F | 110.29 | 129.45 | 152.50 | 64.60 | 75.18 | 81.82 | 93.05 | 111.86 | 126.38 |
|  | ∀ F | 136.49 | 165.86 | 192.45 | 79.69 | 93.48 | 102.72 | **138.90** | 161.20 | 165.63 |
| **GPT-4o** | Van | 67.96 | 74.09 | 80.83 | 57.92 | 59.67 | 62.00 | 105.44 | 121.61 | 132.78 |
|  | F | 125.94 | 145.78 | 166.53 | 83.22 | **88.73** | 93.65 | **124.14** | **139.25** | **154.54** |
|  | ¬ F | 105.40 | 120.77 | 137.27 | 71.54 | 76.56 | 82.59 | 104.14 | 120.23 | 135.47 |
|  | ∀ F | **128.74** | **147.34** | **167.87** | **84.84** | 88.34 | **95.29** | 121.28 | 136.13 | 150.43 |

## C BROADER IMPACTS

In the current research for LLMs, the paper carries a significant broader impact, primarily reflected in the surface learning behavior uncovered, self-concept planning in training-free scenario and behavior correction strategy for post-training proposed to mitigate surface learning. Through mathematical reasoning and grammatical reasoning, as well as the underlying knowledge, we discover the surface learning behavior, which is mainly manifested in three aspects: rote learning, ignoring background knowledge and focusing on answering paradigms. The paper provides research findings from a new perspective for further studying the reliability and knowledge reserve of LLMs.

To further alleviate the surface learning behavior of LLMs, this paper not only provides solutions in the training-free scenario, but also proposes novel behavior correction strategy in post-training process to help the models reduce the reliance on answer paradigms for reasoning. In the training-free scenario, based on the self-concept theory, this paper expands the prompt setting of LLMs in three aspects: goal-setting, planning beforehand, feedback afterward. It can not only help the model improve its reasoning performance, but also offers reasons behind the effectiveness of a series of variants on COT (e.g. CoT-SC, Divide and Conquer, etc.) from the perspective of cognitive science. In the post-training scenario, through the novel behavior correction strategy and the designed indicators, LLMs can effectively activate and schedule the knowledge of base LLMs. This enables LLMs to truly leverage knowledge rather than simply adopting the easy-to-find answer paradigm for reasoning.

Additionally, the broader impact of this paper is manifested in several other aspects: (1) Trustworthy AI Development: By exploring and mitigating surface learning behaviors, our framework advances the pursuit of actual knowledge understanding over answer paradigm. This shift is crucial for deploying LLMs in high-stakes domains (e.g., education, legal analysis) where reliance on surface paradigms could propagate harmful biases or factual errors. (2) Cognitive AI Alignment: The Self-Concept Planning strategy bridges cognitive science with prompt engineering. This cross-disciplinary alignment fosters LLMs that reason with human-compatible logical structures, enhancing their utility in real-world applications. (3) Sustainable LLM Optimization: The Self-Concept Planning in training-free scenario and Behavior Correction strategy in post-training process both alleviate the surface learning behavior of LLM, and promote the ability of LLM to learn knowledge and apply knowledge-a critical step to achieve sustainable learning systems.

## D MORE SETTINGS

### D.1 VANILLA SETTING

```
[SYSTEM] Given definitions of relevant formulas:

Formulas: {}

Please provide the highly relevant formulas to the math question and
provide the answer.

Question: {}

Now you need to output the formula and answer in list format with your
analysis. Your output is a JSON-formatted string that is compressed into
a single line. Here is an example of an output:

example: {}
```

### D.2 $F$ OR $\neg F$ SETTING

```
Given relevant formulas of question:

Formulas: {}

Question: {}
```

```
Now you need to output the answer in list format with your analysis. Your
 output is a JSON-formatted string that is compressed into a single line.
 Here is an example of an output:

example: {}
```

## D.3 ∀F SETTING

```
Given relevant formulas of question:

Formulas: {}

Given irrelevant formulas of question:

Formulas: {}

Question: {}

Now you need to output the answer in list format with your analysis. Your
 output is a JSON-formatted string that is compressed into a single line.
 Here is an example of an output:

example: {}
```

## D.4 VANILLA SETTING IN ENGLISH GRAMMAR TESTS

```
Please read the following English grammar question carefully, give the
most relevant knowledge points to the question and the answer:

Question: {}

Please output the answer in json format. Here is an example output:

example: {}
```

## D.5 SELF-CONCEPT PLANNING

```
Main Task: Figure out the key knowledge relevant to the question.

Subtask: Please provide the highly relevant formula as possible to the
math question and the answer.

Question: {}

Step 1: Rate your understanding from 1 to 10.

Step 2: If the rating is below {}, repeat the Main Task to enhance
understanding.

Step 3: Reassess your understanding after further exploration.

Now you need to output the formula and answer in list format with your
analysis.

Your output is a JSON-formatted string that is compressed into a single
line.

example: {}
```

## D.6 DEFINITIONS OF FORMULAS IN ADD_SUB

```
Formula 1: Addend + Addend = Sum.

Formula 2: Minuend - Subtrahend = Difference.

Formula 3: Adding two numbers with different signs: The result is the
absolute value of the difference between the two numbers; the sign of the
 result is the same as the original sign of the number with the larger
absolute value.

Formula 4: Adding two numbers with the same sign: The result is the sum
of the absolute values of the two numbers; the sign of the result is the
same as the original sign of the two numbers.

Formula 5: Subtracting two numbers with different signs: The result is
the sum of the absolute values of the two numbers; the sign of the result
 is the same as the original sign of the minuend.

Formula 6: Subtracting two numbers with the same sign: The result is the
absolute value of the difference between the two numbers. If the minuend
is greater than the subtrahend, the sign is positive; if the minuend is
less than the subtrahend, the sign is negative.

Formula 7: Decimal points must be aligned when performing addition or
subtraction, meaning that digits in the same place value should be added
or subtracted together.
```

## D.7 DEFINITIONS OF FORMULAS IN MUL_DIV

```
Formula1: Multiplicand * Multiplier = Product.

Formula2: Dividend / Divisor = Quotient.

Formula3: The product or quotient of two negative numbers is a positive
number.

Formula4: The product or quotient of two numbers with different signs is
always a negative number.

Formula5: Any number multiplied by 0 equals 0.

Formula6: Any number divided by 1 equals itself; Any nonzero number
divided by itself equals 1.

Formula7: Multiplying a decimal by 10, 100, 1000, etc. Multiplying a
decimal by these numbers is equivalent to moving the decimal point to the
 right.

Formula8: Dividing a decimal by 10, 100, 1000, etc. Dividing a decimal by
 these numbers is equivalent to moving the decimal point to the left.
```

## D.8 DEFINITIONS OF FORMULAS IN FACTOR

```
Formula2.1: The last digit is an even number (0, 2, 4, 6, 8), indicating
that it is divisible by 2.

Formula3.1: If the sum of the digits is divisible by 3, then the number
itself is divisible by 3.
```

Formula3.2: Subtract the quantity of the digits 2, 5, and 8 in the number from the quantity of the digits 1, 4, and 7 in the number. If result is divisible by 3, then the number itself is divisible by 3.

Formula3.3: Subtracting twice the last digit from the rest gives a multiple of 3. If result is divisible by 3, then the number itself is divisible by 3.

Formula4.1: If the last two digits are divisible by 4, then the number itself is divisible by 4.

Formula4.2: When the tens digit is even, if the ones digit is 0, 4, or 8, then the number is divisible by 4.

Formula4.3: When the tens digit is odd, if the ones digit is 2 or 6, then the number is divisible by 4.

Formula4.4: If the sum of the ones digit and double the tens digit is divisible by 4, then the number itself is divisible by 4.

Formula5.1: If the last digit is 0 or 5, then the number is divisible by 5.

Formula6.1: A number that is divisible by both 2 and 3 is also divisible by 6.

Formula7.1: When grouped in threes from right to left, if the alternating sum is divisible by 7, then the number is divisible by 7.

Formula7.2: If the units digit is multiplied by 2 and subtracted from the other digits, the result must be divisible by 7 for the number to be divisible by 7.

Formula7.3: If the units digit is multiplied by 5 and added to the the other digits, the result must be divisible by 7 for the number to be divisible by 7.

Formula7.4: If the highest digit is multiplied by 3 and added to the next highest digit, replacing the first two digits with the result must yield a number divisible by 7.

Formula7.5: If adding the last two digits to twice the rest, yields a result divisible by 7, then the number is divisible by 7.

Formula7.6: Starting from the last digit and alternating multipliers of 1, 3, 2, -1, -3, -2 (in a repeating cycle), if the sum is divisible by 7, then the number is divisible by 7.

Formula8.1: If the hundreds digit is even and the last two digits form a number divisible by 8, then the entire number is divisible by 8.

Formula8.2: If the hundreds digit is odd, the number obtained by the last two digits must be 4 times an odd number, then the number is divisible by 8.

Formula8.3: If adding the last digit to twice the rest, results in a number divisible by 8, then the number is divisible by 8.

Formula8.4: If the last three digits form a number divisible by 8, then the entire number is divisible by 8.

Formula8.5: If four times the hundreds digit plus two times the tens digit plus the units digit yields a result divisible by 8, then the number is divisible by 8.

Formula9.1: If the sum of the digits is divisible by 9, then the number itself is also divisible by 9.

Formula10.1: The last digit is, the number is divisible by 10.

Formula10.2: If the number is divisible by 2 and by 5, then the number is divisible by 10.

Formula11.1: If the alternating sum of the digits from highest to lowest is divisible by 11, then the number is divisible by 11.

Formula11.2: When grouped in pairs from right to left, if the sum of each group is divisible by 11, then the number is divisible by 11.

Formula11.3: If the last digit subtracted from the rest yields a result divisible by 11, then the number is divisible by 11.

Formula11.4: Add 10 times the last digit to the rest. if the result is divisible by 11, then the number is divisible by 11.

Formula11.5: When the number has an even number of digits, add the first and subtract the last digit from the rest yields a result divisible by 11, then the number is divisible by 11.

Formula11.6: When the number has an odd number of digits, if subtract the first and last digit from the rest yields a result divisible by 11, then the number is divisible by 11.

Formula12.1: A number that is divisible by both 3 and 4 is also divisible by 12.

Formula12.2: If the last digit is subtracted from twice the rest and the result is divisible by 12, then the number is divisible by 12.

Formula13.1: When grouped in threes from right to left, if the alternating sum is divisible by 13, then the number is divisible by 13.

Formula13.2: If subtract the last two digits from four times the rest, yielding a number divisible by 13, then the number is divisible by 13.

Formula13.3: If subtract 9 times the last digit from the rest, and the result is divisible by 13, then the number is divisible by 13.

Formula13.4: If the last digit is multiplied by 9 and the result is subtracted from the rest, yielding a number divisible by 13, then the number is divisible by 13.

Formula14.1: A number that is divisible by both 2 and 7 is also divisible by 14.

Formula14.2: If the last two digits added to twice the rest yield a result divisible by 14, then the number is divisible by 14.

Formula15.1: A number that is divisible by both 3 and 5 is also divisible by 15.

Formula16.1: If the thousands digit is even and the number formed by the last three digits is divisible by 16, then the entire number is divisible by 16.

Formula16.2: If the thousands digit is odd, the number formed by the last three digits is 8 times an odd number, then the entire number is divisible by 16.

Formula16.3: If the last two digits added to four times the rest yield a result divisible by 16, then the number is divisible by 16.

Formula16.4: If the last four digits form a number divisible by 16, then the entire number is divisible by 16.

Formula17.1: If the last digit is multiplied by 5 and subtracted from the rest, yielding a result divisible by 17, then the number is divisible by 17.

Formula17.2: if the result of adding 12 times the last digit to the rest is divisible by 17, then the number is divisible by 17.

Formula17.3: If the last two digits are subtracted from twice the rest and the result is divisible by 17, then the number is divisible by 17.

Formula18.1: A number that is divisible by both 2 and 9 is also divisible by 18.

Formula19.1: If sum of twice the last digit and the rest, yielding a result divisible by 19, then the number is divisible by 19.

Formula19.2: If sum of 4 times the last two digits and the rest, yielding a result divisible by 19, then the number is divisible by 19.

Formula20.1: If the last digit is 0 and the tens digit is even, then the number is divisible by 20.

Formula20.2: If the last two digits form a number divisible by 20, then the entire number is divisible by 20.

Formula20.3: A number that is divisible by both 4 and 5 is also divisible by 20.

Formula21.1: If subtracting twice the last digit from the rest gives a multiple of 21, then the number is divisible by 21.

Formula21.2: If the sum of 19 times the last digit and the rest gives a multiple of 21, then the number is divisible by 21.

Formula21.3: A number that is divisible by both 3 and 7 is also divisible by 21.

Formula22.1: A number that is divisible by both 2 and 11 is also divisible by 22.

Formula23.1: If the last digit is multiplied by 7 and added to the rest, yielding a result divisible by 23, then the number is divisible by 23.

Formula23.2: If the last two digits are multiplied by 3 and added to the rest, yielding a result divisible by 23, then the number is divisible by 23.

Formula23.3: If the last three digits are multiplied by 2 and subtracted from the rest, yielding a result divisible by 23, then the number is divisible by 23.

Formula24.1: A number that is divisible by both 3 and 8 is also divisible by 24.

Formula25.1: If the last two digits form a number that is divisible by 25 (00, 25, 50, or 75), then the entire number is divisible by 25.

Formula26.1: A number that is divisible by both 2 and 13 is also divisible by 26.

```
Formula26.2: If subtracting 5 times the last digit from twice the rest of
 the number gives a multiple of 26, then the entire number is divisible
by 26.

Formula27.1: When grouped in threes, if the sum of each group is
divisible by 27, then the number is divisible by 27.

Formula27.2: If the last digit is multiplied by 8 and subtracted from the
 rest, yielding a result divisible by 27, then the number is divisible by
 27.

Formula27.3: If the last two digits are subtracted from eight times the
rest, yielding a result divisible by 27, then the number is divisible by
27.

Formula28.1: A number that is divisible by both 4 and 7 is also divisible
 by 28.

Formula29.1: If the last digit is multiplied by 3 and added to the rest,
yielding a result divisible by 29, then the number is divisible by 29.

Formula29.2: If the last two digits are multiplied by 9 and added to the
rest, yielding a result divisible by 29, then the number is divisible by
29.

Formula29.3: If the last three digits are multiplied by 2 and subtracted
from the rest, yielding a result divisible by 29, then the number is
divisible by 29.

Formula30.1: A number that is divisible by both 3 and 10 is also
divisible by 30.

Formula31.1: If the last digit is multiplied by 3 and subtracted from the
 rest, yielding a result divisible by 31, then the number is divisible by
 31.
```

### D.9 DEFINITION OF FORMULA IN REMAINDER

```
Remainder = Dividend - Divisor * Quotient; Quotient = Dividend / Divisor
(Quotient retains the integer part).
```

### D.10 DEFINITIONS OF GRAMMAR KNOWLEDGE IN ENGLISH GRAMMAR TESTS

```
Parts of Speech:

[Verb, Non-finite Verb, Modal Verb, Adjective, Adverb, Numeral,
Conjunction, Pronoun, Preposition, Noun, Article]
```

```
Clause:

[Attributive Clause, Appositive Clause, Predicative Clause, Object Clause
, Subject Clause, Cause Adverbial Clause, Adverbial Clause, Time
Adverbial Clause, Condition Adverbial Clause, Result Adverbial Clause,
Concession Adverbial Clause, Purpose Adverbial Clause, Place Adverbial
Clause]
```

```
Tense:

[Past Continuous Tense, Present Continuous Tense, Past Perfect Tense,
Present Perfect Tense, Future-in-the-Past Tense, Simple Past Tense,
Simple Present Tense, Simple Future Tense]
```

```
Sentence Patterns and Structures:

['Subject-Verb Agreement', 'Sentence and Sentence Elements', 'Exclamatory
 Sentence', 'Passive Voice', 'Subjunctive Mood', 'Imperative Sentence', '
Phrase and Idiom', 'Situational Interaction']
```

