# OpenReview forum: "Beneath the Surface: Exposing and Mitigating Surface Learning in Large Language Models"
_ICLR.cc/2026/Conference — Submitted to ICLR 2026_

### Official Review · Reviewer_4jH5 · 2025-10-17

**Soundness:** 3
**Presentation:** 3
**Contribution:** 2
**Rating:** 4
**Confidence:** 4

**Summary:**

This paper investigates surface learning behavior in Large Language Models (LLMs). The authors design clever experiments by providing correct/incorrect formulas or concepts alongside mathematical reasoning and English grammar questions. They identify three types of surface learning: (1) rote learning—models memorize formulas without understanding the underlying concepts, (2) ignoring background knowledge—providing relevant or irrelevant formulas doesn't necessarily improve or harm performance, and (3) focusing on answer patterns—models learn spurious correlations between concepts and solutions. To address this, the authors propose a long-term strategy including self-concept planning (training-free) and behavior correction (post-training).

**Strengths:**

1. Experimental design: The idea of testing model understanding by providing correct/incorrect formulas is innovative and insightful. This controlled setup effectively reveals whether models utilize the formulas or concept.
2. Well-written with good motivation: The paper uses concrete examples to illustrate surface learning phenomena. The narrative is clear, and the problem is well-motivated. This is a worthwhile research direction.
3. Comprehensive evaluation: The paper evaluates multiple open-source, closed-source, and reasoning models, providing broad coverage.

**Weaknesses:**

1. Incomplete Problem Framing:
  - This paper aims to address surface learning—where models possess the relevant knowledge but fail to apply it correctly. However, the experiments only use multiple-choice and true/false questions. In these question types, models can obtain correct rewards through incorrect reasoning, a phenomenon known as shortcut learning or reward hacking. Yet surface learning should not be limited to shortcut learning alone. For example, in open-ended math problems where the solution space is vast and models cannot simply guess the correct answer, there still exists the scenario where models know the formulas but cannot solve the problems. We suggest incorporating open-ended question experiments to capture a more complete picture of surface learning beyond shortcut learning.
2. Missing critical implementation details:
  - The multi-turn dialogue implementation is not explained in either the main text or the appendix, undermining the credibility of Section 5.2.2's conclusions
  - Inverse-BC is not clearly defined. If it simply inverts the reward ranking within groups, this baseline seems both odd and poorly motivated

3. Counter-intuitive reward design: The appendix mentions increasing rewards for incorrect outputs, claiming this "indicates the model doesn't use answer paradigms to steal scores." This logic is questionable:
  - The model may be attempting to game the reward but failing
  - RL methods like GRPO require accurate rewards; rewarding incorrect outputs is highly counter-intuitive
  - This claim needs much more analysis.

4. Fragile conclusions:
  - Figures 6 and 7 show tiny differences (0.01-0.02 or smaller) between methods. Given GRPO's inherent training variance, these differences are not convincing enough
  - Conclusion 3 is based on non-F (providing the wrong formula) hurting performance on English questions, but non-F also performs worst on math Add problems. The relationship between F/non-F and performance needs more systematic analysis

5. No downstream validation: Experiments are limited to the constructed ME-Test suite. There's no validation on standard benchmarks (MATH, GSM8K, etc.). It's unclear whether these toy experiment findings generalize to real applications.

Minor Weaknesses

6. Key details buried in appendix: The reward indicator I is only explained in the appendix. Core methodological details should be clearly stated in the main text.

7. Scattered conclusions: The three surface learning behaviors and their mitigation strategies show inconsistent effectiveness across settings, lacking a unified explanatory framework.

**Questions:**

see Weaknesses

---

> ### Author Response · Authors · 2025-11-21
> **Response to Reviewer 4jH5 (1)**
>
> We thank the reviewer for the thoughtful and detailed comments. We are pleased that the reviewer considers our innovative and insightful designs, well-written with good motivation! We appreciate the opportunity to address the concerns here.
>
> ---
>
> **Comment1**: Incomplete Problem Framing.
>
> **Answer**: Thanks for your question. We want to clarify two misunderstandings. Regarding the first misunderstanding:
> > the experiments only use multiple-choice and true/false questions. We suggest incorporating open-ended question experiments to capture a more complete picture of surface learning beyond shortcut learning.
>
> The DeepMind Mathematics dataset, which forms the foundation of the mathematical problems in our ME-Test suite, contains multiple problem types with varying formats. Among these, **only the Factor subset consists of true/false questions**—where the model must judge whether a given number is a factor of another. In contrast, **the remaining subsets (Add_Sub, Mul_Div, Rem) are open-ended**, requiring the model to generate numerical answers without predefined options or binary judgments. This structural variation is inherent to the dataset design and reflects real-world mathematical problem formats, as an example in Figure 13. Importantly, the "true/false" format in the Factor subset arises naturally from the nature of factorization tasks—not as an artificial constraint. Similarly, in English QA dataset (e.g., those referenced in lines 155–157), **open-ended formats such as fill-in-the-blank are standard and widely accepted** alongside multiple-choice questions. The DeepMind Mathematics dataset is a **well-established, community-recognized benchmark** for evaluating mathematical reasoning. For reproducibility, we have included an anonymous, downloadable version of our constructed ME-Test suite, ensuring full transparency and accessibility.
>
>
> Regarding the second misunderstanding:
> > there still exists the scenario where models know the formulas but cannot solve the problems.
>
> This is precisely the main focus of the paper: The surface learning behavior indicates that although the models seem to know the formulas and strategies required to solve specific types of problems on the surface, they do not truly comprehend the essence of these concepts and fails in solving problems, resulting in surface-level short-term benefits rather than in-depth learning.
>
> ---
>
> **Comment2**: Missing critical implementation details.
>
> **Answer**:
> - In lines 260-261, we provided a detailed description of the multi-turn dialogue setup. **Multi-turn dialogues can be constructed by sequentially appending messages with the "assistant" role, with a max generation length of 1024 tokens per turn and default temperature/top-p settings**.
>
> - To validate the design of the self-cognition metric I and the BC strategy, **we introduce Inverse-BC as a comparable baseline**. Since models more strongly and densely concentrate on features that are simpler to compute or learned first **as illustrated in Section 4.3.2**, we construct BC strategy to prioritize learning ​uncertain and ​oscillating samples through self-cognition metric I, which defines the degree of uncertainty of LLM on the samples. In contrast, ​deterministic samples (low std, high $I^r$, i.e., always learned or never learned) encourage surface learning. Prioritizing them too early (Inverse-BC strategy) can lead the model to rely on answer paradigms, hindering true reasoning development.
>
>  ---
>
> **Comment3**: Counter-intuitive reward design.
>
> **Answer**: Thanks for your question. Regarding the misunderstanding "increasing rewards for incorrect outputs", the whole paper does not involve reward modification or "rewarding incorrect outputs". **All rewards are computed via a fixed, rule-based function (VR task)**. There is no aspect of "game the reward" or "rewarding incorrect outputs" in the paper. **The self-cognition metric $I^r$ is designed to identify and prioritize learning ​uncertain and ​oscillating samples—those with scattered rewards (high std), low overall performance (low mean), or large reward fluctuations (low min).**

---

> > ### Author Response · Authors · 2025-11-21
> > **Response to Reviewer 4jH5 (2)**
> >
> > **Comment4**: Fragile conclusions.
> >
> > **Answer**:
> > -  First, we select the best-performing checkpoints from the baselines and the Behavior Correction (BC) strategy. On Qwen-7B, **BC achieves up to a 23.5% improvement on the remainder subset over training on random samples (RFT)**, with consistent gains across math and English tasks. **Crucially, when formulas are provided (|F), BC shows superior performance, indicating deeper comprehension and mitigating surface learning**. Second, as shown in **Appendix B.5**, we provide a multi-metric comparison of three reinforcement fine-tuned models under three training mechanisms (w/o, w/-, w/). The results demonstrate that **BC promotes more diverse and robust reasoning processes**, confirming its effectiveness in enhancing reasoning quality and mitigating surface learning behaviors.
> >
> > - As the reviewer 4jH5 noted, irrelevant formulas hurt LLM performance on the add_sub set. However, **even with relevant formulas (F setting), we observe no improvement compared to English questions—indicating that models do not leverage the formulas meaningfully**. If they had learned a spurious correlation between "addition/subtraction" formulas and correct answers, performance would improve in the F setting. **The absence of such gain shows their behavior cannot be explained by Conclusion 3 alone, and thus we avoid oversimplifying the results for this subset.**
> >
> > ---
> >
> > **Comment5**: No downstream validation.
> >
> > **Answer**: ME-Test is grounded in a standard benchmark: The mathematical problems in the ME-Test suite are derived from the **DeepMind Mathematics Dataset, which is widely recognized as a standard benchmark for evaluating mathematical reasoning in language models**. Besides, the addition of formulas and knowledge in ME-Test suite provide a controlled experimental setting to diagnose and isolate surface learning behaviors of LLMs.
> >
> > ---
> > **Comment6**: Key details buried in appendix.
> >
> > **Answer**: Thanks for your suggestion. In Section 4.3.2, we provided a detailed explanation of self-cognition I. In Section B.2, we further elaborated on the specific circumstances covered by Formula (1) and the problems it addressed.
> >
> > ---
> >
> > **Comment7**: Scattered conclusions.
> >
> > **Answer**: Thanks for this insightful comment. We agree that a unified perspective enhances interpretability, and in fact, our work is grounded in a **coherent theoretical foundation from educational psychology**: the concept of surface learning behavior. In human education, surface learning refers to strategies such as rote memorization, cue-based guessing, or reliance on superficial patterns - rather than deep conceptual understanding. Our paper **systematically adapts this well-established framework to LLMs**, identifying three representative behavior of surface learning.
> >
> > Rather than proposing ad hoc fixes, our mitigation strategies are **derived from the same psychological principles** that explain the behaviors:
> >
> > - In **training-free settings**, we draw on **Self-Concept Theory** and educational interventions for human surface learners. This theory suggests that encouraging goal-setting, planning, and feedback forward helps shift learners from surface to deep learning.
> >
> > - In **post-training settings**, we address the observed phenomenon in **Takeaway 3**. To mitigate this, based on the research that models more strongly and densely concentrate on features that are simpler to compute or learned first. We propose BC strategy with self-cognition to mitigate surface learning of LLMs in training process.
> >
> > We will improve our work based on all the constructive comments, and we are grateful for any additional feedback and suggestions.

---

> ### Comment · Reviewer_4jH5 · 2025-11-28
>
> Thank you for the response. It answered some of my concerns. However, I still have some confusion:
> 1. The precise format of the multi-turn exchange remains an unresolved issue. It would be better to provide some examples in the writing.
> 2. While the BC is intuitively appealing, its verification is currently limited to a toy dataset with simplistic patterns. Downstream testing refers to validating the method on more common mathematical datasets to justify it.
> If you provide downstream evaluation experiments, I will raise my score. For now, I plan to lower my confidence to 3 : ).
>
> (Why can't I change the confidence of my review opinions? Does AC know?)

---

> ### Author Response · Authors · 2025-12-03
> **Response to Reviewer 4jH5**
>
> Thanks for Reviewer 4jH5’s suggestion. we are committed to thoroughly addressing the concerns.
>
> **1. multi-turn exchange.**
>
> The reviewer appears to have misunderstood the multi-turn dialogue setup in the paper. **Our approach does not involve a multi-agent debate or iterative exchanges between agents**. As stated in lines 37–39: “*Since problem-solving typically follows the principle of progressive learning—from simple to complex—grounded in Cognitive Load Theory (Sweller, 2011), we construct a sequence of questions with increasing difficulty, enabling the model to engage in continuous problem-solving by leveraging the multi-turn dialogue history.*” Furthermore, the left panel of Figure 1 clearly illustrates this design: **the model does not answer each question in isolation but instead uses prior dialogue context (e.g., its previous responses) to inform subsequent answers.**
>
> **2. BC on more common mathematical datasets.**
>
> We appreciate the reviewer's acknowledgment of the BC strategy as intuitively appealing. For datasets like GSM8K—where problems involve **a mix of arithmetic operations and require semantic understanding**—it is inherently difficult to precisely delineate relevant from irrelevant knowledge. The surface patterns exhibited by models may extend beyond the **answer paradigms discussed in this paper**, and verifying such surface patterns typically requires carefully designed prompts (e.g., explicitly prompting for formula insertion). Therefore, the generalizability of the BC method cannot be established merely by applying it to a large number of datasets.
>
> **In the absence of fine-grained knowledge annotations**, we conducted a comparison between the BC strategy and baseline methods using LLaMA3.1-3B on GSM8K, which demonstrates the effectiveness of BC. However, without explicit knowledge labels, **it remains challenging to determine whether the BC strategy genuinely helps the model acquire deeper understanding of the underlying knowledge. The construction of ME-Test effectively addresses this limitation**. **ME-Test is not merely a toy dataset; it includes complex mathematical problems along with both relevant and irrelevant formulas, thereby aligning thoroughly and effectively with the core motivation of our work**.
>
> The comparison between the baselines and the BC strategy is based on the results generated by their respective best-performing checkpoints.
> |          | RFT   | Inverse-BC | BC    |
> |----------|-------|------------|-------|
> | LLaMA3.1-3B | 0.710 | 0.526      | **0.763** |
>
> We sincerely thank Reviewer 4jH5 for the thoughtful and constructive engagement with our work and for the positive acknowledgment of our paper!

---

### Official Review · Reviewer_P4ic · 2025-11-01

**Soundness:** 3
**Presentation:** 3
**Contribution:** 3
**Rating:** 6
**Confidence:** 3

**Summary:**

The paper studies surface learning in LLMs. They define it as cases where models recall formulas or patterns but fail to deeply understand and apply them. The authors introduce ME-Test that tests an LLM with progressive difficulty and exposure to controlled formula. They show that on this task, the model uses surface level heuristics. To mitigate, they propose 2 strategies: prompting for self-Concept planning and post-training  with a behavior-correction sampling strategy. Through extensive experiments on LLMs, they show that surface level learning can be mitigated in LLMs.

**Strengths:**

The paper clearly identifies and formalizes surface learning  in LLMs. By proposing a controlled testing framework, the authors show that LLMs simply recall symbolic rules but fail to apply them robustly to the task at hand. To mitigate, the authors propose 2 intuitive strategies, following human pedagogy. First, they propose to prompt the model to Self-Concept Planning, that asks the model to generate a plan before solving the question. The second strategy aims to modify sampling in post-training such that the model doesn't learn surface level features. Overall, the authors show a timely evaluation of LLMs and propose careful mitigation strategies to solve the problem.

**Weaknesses:**

The primary concern is equating surface level learning with the degradation in performance at more difficult questions. The failure of the model could be because of different reasons, e.g. hallucinations when generating longer answers. How would the authors disentangle issues with long form generation with surface level learning? The authors can present the average response length across the 3 settings and try to disentangle length to strengthen their argument.

Second, instead of simply asking the model about which concepts are relevant in prompt style mitigation, could the authors ask the model to reason about all concepts, and reason about irrelevant concepts as well. Some of $2$-way competition could improve the performance of prompting strategies.

How much does the structure of prompt affect the experiment results? For example, do the authors observe major differences in performance, depending on how the formulas are ordered in the prompt?

**Questions:**

Please check above for my questions.

---

> ### Author Response · Authors · 2025-11-21
> **Response to Reviewer P4ic (1)**
>
> We thank the reviewer for the thoughtful and detailed comments. We’re glad the reviewer found our identification and formalization of surface learning clear and intuitive, and appreciate the chance to address the concerns here.
>
> ---
>
> **Comment1**: The authors can present the average response length across the 3 settings and try to disentangle length to strengthen their argument.
>
> **Answer**: Thanks for your valuable question! Taking into account the output tokens, we list the output length of different LLMs under different prompt settings in **Appendix B.8. It can be seen that there is no obvious relationship between model performance and tokens length. The best/worst model performance is often not the one with the longest/shortest length**.
>
> ---
> **Comment2**: Second, instead of simply asking the model about which concepts are relevant in prompt style mitigation, could the authors ask the model to reason about all concepts, and reason about irrelevant concepts as well. Some of  -way competition could improve the performance of prompting strategies.
>
> **Answer**: Thanks for the valuable suggestion! We adjust the subtask prompts of LLMs under the Self-Concept Planning (SC-P) strategy as follows:
> > Subtask: Please provide the highly relevant formula as possible to the math question and the answer.
>
> > Subtask: Clearly explain which formulas are relevant and which are not, along with the reasons for this, and finally apply the highly relevant formulas to solve the problem and provide the answer.
>
> The performance of Qwen-14B under different strategies in mathematical subset (a, f represent performance on **a**nswering and predicting **f**ormulas).
>
> | Method | Easy-a | Easy-f | Medium-a | Medium-f | Hard-a | Hard-f |
> |--------|--------|--------|----------|----------|--------|--------|
> | Vanilla | 85.04 | 81.18 | 77.34 | 84.20 | 76.74 | 82.74 |
> | SC-P | 89.36 | **86.64** | 80.64 | **84.64** | 79.12 | **84.48** |
> | SC-P+¬ F | **90.16** | 73.76 | **84.13** | 68.20 | **81.06** | 66.50 |
>
>
> The performance of LLaMA-70B under different strategies in mathematical subset.
>
> | Method | Easy-a | Easy-f | Medium-a | Medium-f | Hard-a | Hard-f |
> |--------|--------|--------|----------|----------|--------|--------|
> | Vanilla | 76.42 | 74.02 | 72.62 | 82.46 | 72.06 | 83.88 |
> | SC-P | **77.02** | **80.84** | **73.14** | **84.06** | 72.20 | **84.44** |
> | SC-P+¬ F | 76.63 | 79.36 | 72.56 | 76.26 | **73.90** | 78.23 |
>
>
> The above experimental results show that asking the model to reason about all concepts can enhance its performance on certain subsets to some extent. However, **it is notable that when reasoning irrelevant formulas, Qwen-14B frequently misclassifies them as relevant alongside truly relevant ones in the final prediction stage**. This results in a significant decline in the performance of formula predictions, revealing a clear inconsistency between its internal reasoning process and the output results. We will incorporate these findings into the paper to emphasize reasoning consistency in LLMs.
>
> ---

---

> > ### Author Response · Authors · 2025-11-21
> > **Response to Reviewer P4ic (2)**
> >
> > ---
> >
> > **Comment3**: How much does the structure of prompt affect the experiment results? For example, do the authors observe major differences in performance, depending on how the formulas are ordered in the prompt?
> >
> > **Answer**: Thanks for your critical suggestion! We randomly changed the positions of the formulas and evaluate 5000 mathematical questions for each subset and each setting. The performance comparison of various LLMs in multi-turn evaluation with randomly ordered formulas are as follows ("order" and "random" denote the settings of sequentially arranged formulas and randomly shuffled formula order, respectively):
> >
> >
> > **The performance comparison of LLaMA3.1-8B.**
> >
> > | Model | Setting | Add\_Sub (Easy) | Add\_Sub (Med) | Add\_Sub (Hard) | Mul\_Div (Easy) | Mul\_Div (Med) | Mul\_Div (Hard) | Factor (Easy) | Factor (Med) | Factor (Hard) |
> > |-------|---------|------------------|----------------|------------------|------------------|----------------|------------------|----------------|---------------|----------------|
> > | order | F | 0.382 | 0.301 | 0.200 | 0.594 | 0.327 | 0.183 | 0.756 | 0.712 | 0.695 |
> > | order | ¬F | 0.108 | 0.087 | 0.056 | 0.362 | 0.259 | 0.161 | 0.751 | 0.661 | 0.633 |
> > | order | ∀F | 0.386 | 0.307 | 0.209 | 0.570 | 0.319 | 0.184 | 0.740 | 0.694 | 0.667 |
> > | random | F | 0.335 | 0.271 | 0.188 | 0.216 | 0.133 | 0.077 | 0.759 | 0.708 | 0.692 |
> > | random | ¬F | 0.112 | 0.101 | 0.065 | 0.150 | 0.125 | 0.069 | 0.623 | 0.577 | 0.568 |
> > | random | ∀F | 0.324 | 0.268 | 0.184 | 0.211 | 0.127 | 0.071 | 0.707 | 0.689 | 0.660 |
> >
> > ---
> >
> > **The performance comparison of  LLaMA3.1-70B.**
> >
> > | Model | Setting | Add\_Sub (Easy) | Add\_Sub (Med) | Add\_Sub (Hard) | Mul\_Div (Easy) | Mul\_Div (Med) | Mul\_Div (Hard) | Factor (Easy) | Factor (Med) | Factor (Hard) |
> > |-------|---------|------------------|----------------|------------------|------------------|----------------|------------------|----------------|---------------|----------------|
> > | order | F | 0.859 | 0.794 | 0.626 | 0.812 | 0.578 | 0.336 | 0.857 | 0.742 | 0.696 |
> > | order | ¬F | 0.835 | 0.796 | 0.659 | 0.831 | 0.575 | 0.344 | 0.594 | 0.486 | 0.456 |
> > | order | ∀F | 0.856 | 0.780 | 0.614 | 0.817 | 0.571 | 0.324 | 0.503 | 0.411 | 0.396 |
> > | random | F | 0.845 | 0.791 | 0.627 | 0.889 | 0.608 | 0.312 | 0.862 | 0.735 | 0.701 |
> > | random | ¬F | 0.806 | 0.786 | 0.663 | 0.911 | 0.608 | 0.321 | 0.611 | 0.450 | 0.426 |
> > | random | ∀F | 0.854 | 0.775 | 0.617 | 0.883 | 0.581 | 0.291 | 0.443 | 0.423 | 0.387 |
> >
> > ---
> >
> > **The performance comparison of Qwen2.5-14B.**
> >
> > | Model | Setting | Add\_Sub (Easy) | Add\_Sub (Med) | Add\_Sub (Hard) | Mul\_Div (Easy) | Mul\_Div (Med) | Mul\_Div (Hard) | Factor (Easy) | Factor (Med) | Factor (Hard) |
> > |-------|---------|------------------|----------------|------------------|------------------|----------------|------------------|----------------|---------------|----------------|
> > | order | F | 0.911 | 0.907 | 0.862 | 0.845 | 0.698 | 0.519 | 0.894 | 0.859 | 0.828 |
> > | order | ¬F | 0.880 | 0.861 | 0.823 | 0.850 | 0.697 | 0.523 | 0.962 | 0.892 | 0.853 |
> > | order | ∀F | 0.900 | 0.890 | 0.853 | 0.852 | 0.698 | 0.515 | 0.924 | 0.853 | 0.840 |
> > | random | F | 0.916 | 0.904 | 0.867 | 0.745 | 0.627 | 0.463 | 0.951 | 0.902 | 0.854 |
> > | random | ¬F | 0.878 | 0.859 | 0.814 | 0.724 | 0.610 | 0.452 | 0.962 | 0.894 | 0.857 |
> > | random | ∀F | 0.900 | 0.896 | 0.865 | 0.745 | 0.615 | 0.461 | 0.962 | 0.911 | 0.863 |
> >
> > ---
> >
> > **The performance comparison of GPT-4o.**
> >
> > | Model | Setting | Add\_Sub (Easy) | Add\_Sub (Med) | Add\_Sub (Hard) | Mul\_Div (Easy) | Mul\_Div (Med) | Mul\_Div (Hard) | Factor (Easy) | Factor (Med) | Factor (Hard) |
> > |-------|---------|------------------|----------------|------------------|------------------|----------------|------------------|----------------|---------------|----------------|
> > | order | F | 0.882 | 0.880 | 0.883 | 0.951 | 0.849 | 0.611 | 0.904 | 0.854 | 0.859 |
> > | order | ¬F | 0.743 | 0.780 | 0.770 | 0.950 | 0.856 | 0.616 | 0.909 | 0.867 | 0.857 |
> > | order | ∀F | 0.799 | 0.793 | 0.796 | 0.962 | 0.863 | 0.616 | 0.826 | 0.783 | 0.780 |
> > | random | F | 0.907 | 0.904 | 0.901 | 0.976 | 0.881 | 0.623 | 0.916 | 0.869 | 0.855 |
> > | random | ¬F | 0.842 | 0.903 | 0.900 | 0.969 | 0.869 | 0.630 | 0.907 | 0.880 | 0.857 |
> > | random | ∀F | 0.901 | 0.891 | 0.903 | 0.968 | 0.871 | 0.632 | 0.857 | 0.806 | 0.792 |
> >
> > ---
> >
> > Overall, **randomly permuting the order of formulas has minimal impact on model performance and does not affect the findings of surface learning**. However, on the Mul_Div subset, LLaMA-8B and Qwen-14B perform worse under randomized formula-order prompts, revealing their vulnerability—particularly when the models lack genuine mastery of the underlying mathematical knowledge.
> >
> > We will improve our work based on all the constructive comments, and we are grateful for any additional feedback and suggestions.

---

### Official Review · Reviewer_ga6M · 2025-11-02

**Soundness:** 2
**Presentation:** 3
**Contribution:** 2
**Rating:** 4
**Confidence:** 4

**Summary:**

The paper studies a phenomenon the authors call surface learning: LLMs achieving high accuracy on benchmarks by exploiting superficial correlations rather than engaging in deeper reasoning. They propose ME-Test, a probing framework that pairs original questions with minimally-edited variants where surface cues are altered but core semantics remain. Performance drops under these variants are used to quantify surface learning. To mitigate the issue, the paper introduces two interventions: thinking-path correction (prompting models to revise or reflect on earlier reasoning) and behavior correction (explicitly penalizing shallow patterns during generation). Experiments on GSM8K, StrategyQA, and BBH tasks show large performance drops under ME-Test for several open and closed models; the proposed mitigations yield partial recovery.

**Strengths:**

1. ME-Test is simple, reproducible, and architecture-agnostic: minimal edits to input help separate form-matching from semantic reasoning.

2. The performance gaps are systematically measured and provide clear empirical evidence of shortcut reliance.

3. Mitigation strategies are lightweight (prompt-level) and do not require retraining.

**Weaknesses:**

1. The concept of “surface learning” is not clearly distinguished from existing notions like pattern matching, or spurious correlations. The paper introduces a new term but does not establish conceptual novelty.

2. ME-Test edits sometimes oversimplify meaning preservation, and no human validation is provided to confirm that all minimal edits preserve problem semantics fully.

3. Mitigation methods resemble prompt engineering and chain-of-thought reflection; they are not fundamentally novel, and their effectiveness varies across tasks.

4. Improvements after mitigation are moderate and do not eliminate the gap; no analysis is provided on failure cases or when mitigation backfires.

5. No examination of whether ME-Test correlates with real-world robustness or downstream utility; unclear if this is just another adversarial dataset.

**Questions:**

1. How is “surface learning” fundamentally different from shortcut learning or pattern exploitation described extensively in prior literature?

2. How do you ensure that ME-Test edits do not unintentionally change semantics or difficulty? Was human validation performed?

3. Does ME-Test correlate with other robustness metrics or real user-facing failures?

4. Can the mitigation methods cause degradation in standard accuracy or introduce verbosity without improving reasoning quality?

5. Do models with explicit training for chain-of-thought (e.g., R1, DeepSeek-R1) still show the same pattern?

---

> ### Author Response · Authors · 2025-11-21
> **Response to Reviewer ga6M (1)**
>
> ​We thank the reviewer’s time and feedback. We are glad that the ME-Test construction, clear empirical evidence, mitigation strategies have been recognized. However, we need to clarify some factual errors in the summary:
>
> ```
> The paper studies a phenomenon the authors call surface learning: LLMs achieving high accuracy on benchmarks by exploiting superficial correlations rather than engaging in deeper reasoning. They propose ME-Test, a probing framework that pairs original questions with minimally-edited variants where surface cues are altered but core semantics remain. Performance drops under these variants are used to quantify surface learning. To mitigate the issue, the paper introduces two interventions: thinking-path correction (prompting models to revise or reflect on earlier reasoning) and behavior correction (explicitly penalizing shallow patterns during generation). Experiments on GSM8K, StrategyQA, and BBH tasks show large performance drops under ME-Test for several open and closed models; the proposed mitigations yield partial recovery.
> ```
> ​
> **​ME-Test is not a probing framework** that pairs original questions with minimally-edited variants. **No edited variants were involved**. ME-Test suite is consist of a benchmark-Deepmind math dataset and English QA dataset with math formulas and grammar knowledge, which helps to evaluate surface learning. Besides, **​no GSM8K, StrategyQA, and BBH tasks are adopted in the paper**​. We appreciate the opportunity to address the concerns here.
>
> ---
>
> **Comment1**: The concept of "surface learning" is not clearly distinguished from existing notions like pattern matching, or spurious correlations. The paper introduces a new term but does not establish conceptual novelty. **& Q1**: How is "surface learning" fundamentally different from shortcut learning or pattern exploitation described extensively in prior literature?
>
> **Answer**: Thanks for your question. As illustrated in **Appendix B.1, we clearly introduce the comparison of shortcut learning and surface learning**. Although shortcut learning is well defined as model rely on non-robust features or spurious correlations between certain features and labels in classification task, shortcut learning lacks a clear definition in generation tasks. In this paper, we comprehensively evaluate and define Surface Learning behavior in LLMs—a typical and respective form of shortcut learning behavior—where models exhibit three representative behaviors indicating they have learned surface associations between questions and solution paradigms, without genuinely understanding the underlying knowledge or solving the tasks.
>
> ---
>
> **Comment2**: How do you ensure that ME-Test edits do not unintentionally change semantics or difficulty? Was human validation performed? **& Comment5**: No examination of whether ME-Test correlates with real-world robustness or downstream utility; unclear if this is just another adversarial dataset. **& Q2**: ME-Test edits sometimes oversimplify meaning preservation, and no human validation is provided to confirm that all minimal edits preserve problem semantics fully. **& Q3**: Does ME-Test correlate with other robustness metrics or real user-facing failures?
>
> **Answer**: **ME-Test suite is totally not concerns with edits, semantics change, adversarial dataset, or minimal edits**. ME-Test suite includes Mathematical and English grammar examinations, where each question is equipped with relevant knowledge (formulas or grammar knowledge). It helps comprehensively identify and formalize surface learning behavior of LLMs.
>
> ---
>
> **Comment3**: Mitigation methods resemble prompt engineering and chain-of-thought reflection; they are not fundamentally novel, and their effectiveness varies across tasks.
>
> **Answer**: The mitigation methods are not prompt engineering and reflection. They mainly consist of two strategies, both of which are based on human cognition. These strategies mainly involve targeted mitigation of surface learning in scenarios where no training is required and in post-training scenarios.
>
> - In training-free scenario, inspired by Self-Concept theory, LLMs are prompted with **goal-setting** and **planning beforehand** as well as **feedback afterward** to improve the ability in reasoning process. The three core steps involve rigorous cognitive processes that are directly applied to alleviate surface learning.
>
> - In post-training process (RFT/SFT), the behavior correction strategy is proposed to re-rank samples and priorities training  on uncertain samples based on the designed self-cognition indicators I of LLMs.

---

> > ### Author Response · Authors · 2025-11-21
> > **Response to Reviewer ga6M (2)**
> >
> > **Comment4**: Improvements after mitigation are moderate and do not eliminate the gap; no analysis is provided on failure cases or when mitigation backfires.
> >
> > **Answer**: **The Self-Concept Planning strategy significantly improves LLM performance on both math and English tasks, promoting deeper understanding of the underlying knowledge**. Similarly, the Behavior Correction (BC) strategy enhances performance across math and English tasks, **with consistent gains when formulas are provided (|F)**, indicating deeper knowledge comprehension other than surface learning. In **Appendix B.5**, the multi-metric comparisons demonstrate its effectiveness in enhancing reasoning quality and mitigating surface learning behavior. In **Figures 11 and 12**, we present case studies comparing different RFT methods, illustrating how surface learning causes model failure through reliance on easy-to-find paradigms, and highlighting the effectiveness of BC strategy in promoting more robust reasoning.
> >
> > ---
> >
> > **Q4**: Can the mitigation methods cause degradation in standard accuracy or introduce verbosity without improving reasoning quality?
> >
> > **Answer**: **As shown in Section 5.2.2 and Appendix B.5, BC strategy not only improves accuracy but also enhances reasoning quality**, encouraging the model to explore more diverse and robust reasoning paths.
> >
> >
> > ---
> >
> > **Q5**: Do models with explicit training for chain-of-thought (e.g., R1, DeepSeek-R1) still show the same pattern?
> >
> > **Answer**: **As shown in Table 1 and Section 5.2.1, reasoning-focused LLMs such as DeepSeek-R1 and o3-mini exhibit the same surface learning behavior**, confirming that this phenomenon is also a general characteristic of current reasoning models.
> >
> > We will improve our work based on all the constructive comments, and we are grateful for any additional feedback and suggestions.

---

### Author Response · Authors · 2025-12-03

**We sincerely appreciate the time and effort all reviewers made in evaluating our work!** We are also delighted that **reviewers recognize the significance of our research and the value of our findings**:
* The identification and formalization of surface learning in LLMs represent a **promising and worthwhile research direction**. (Reviewers P4ic, 4jH5)
* **Comprehensive and timely evaluation**. (Reviewers P4ic, 4jH5)
* **Intuitive and effective strategies** to mitigate surface learning. (Reviewers P4ic, 4jH5)
* **Innovative and insightful experimental design**. (Reviewers P4ic, 4jH5)

In response to the reviewers’ feedback and constructive suggestions, **we address their concerns by providing additional experiments that thoroughly resolve these issues**:

* The experiments require the model to **reason about all concepts**—including those that are irrelevant. (Reviewer P4ic)
* The evaluation involved **randomly shuffling the positions of the formulas**. (Reviewer P4ic)
* **Performance comparison** between the BC strategy and baseline methods on GSM8K. (Reviewer 4jH5)

Moreover, **we would like to clarify several factual errors raised by Reviewer ga6M**:

* ME-Test suite is **totally not concerns with** edits, semantics change, adversarial dataset, or minimal edits.
* **No GSM8K, StrategyQA, and BBH tasks** are adopted in the paper.
* The mitigation methods are **not prompt engineering and reflection**. The two strategies mainly involve targeted mitigation of surface learning in scenarios where in training-free and post-training scenarios.

We will continue to incorporate reviewers' feedback and improve the paper. Thanks for all constructive suggestions!

---

### Meta-Review · Area_Chair_koYC · 2026-01-04

**Summary:**

There are a few concerns raised by the reviewers:

1. The most prominent one is that there are a lot of confounding factors in the experiment---all reviewers have expressed this concern in some form. I join the reviewers in agreeing that this is a valid concern. And the proposed method does not fundamentally resolve the issue.

2. The remaining concerns, such as missing details and clarity issues, are addressed by the authors in the rebuttal.

**Reviewer Concerns:**

As mentioned above, I think Concern 1 remains and is unlikely to be resolved under the current framing, although other concerns are fairly addressed by the authors.

**Reviewer Scores:**

I don't think any reviewer will change their rating, as the authors' responses did not touch the fundamental weakness raised by the reviewers.

---

### Decision · Program_Chairs · 2026-01-26

Reject